# The Delta Learning Hypothesis:
# Preference Tuning on Weak Data can Yield Strong Gains

**Scott Geng**♣    **Hamish Ivison**♣♠    **Chun-Liang Li**♣    **Maarten Sap**♡    **Jerry Li**♣
**Ranjay Krishna**♣♠    **Pang Wei Koh**♣♠
♣University of Washington    ♠Allen Institute for AI    ♡Carnegie Mellon University
sgeng@cs.washington.edu    ⬡ GitHub Repo

## Abstract

Improvements in language models are often driven by improving the quality of the data we train them on, which can be limiting when strong supervision is scarce. In this work, we show that paired preference data consisting of individually weak data points can enable gains beyond the strength of each individual data point. We formulate the **delta learning hypothesis** to explain this phenomenon, positing that the relative quality *delta* between points suffices to drive learning via preference tuning—even when supervised finetuning on the weak data hurts. We validate our hypothesis in controlled experiments and at scale, where we post-train 8B models on preference data generated by pairing a small 3B model's responses with outputs from an even smaller 1.5B model to create a meaningful delta. Strikingly, on a standard 11-benchmark evaluation suite (MATH, MMLU, etc.), our simple recipe matches the performance of Tülu 3, a state-of-the-art open model tuned from the same base model while relying on much stronger supervisors (e.g., GPT-4o). Thus, delta learning enables simpler and cheaper open recipes for state-of-the-art post-training. To better understand delta learning, we prove in logistic regression that the performance gap between two weak teacher models provides useful signal for improving a stronger student. Overall, our work shows that models can learn surprisingly well from paired data that might typically be considered weak.

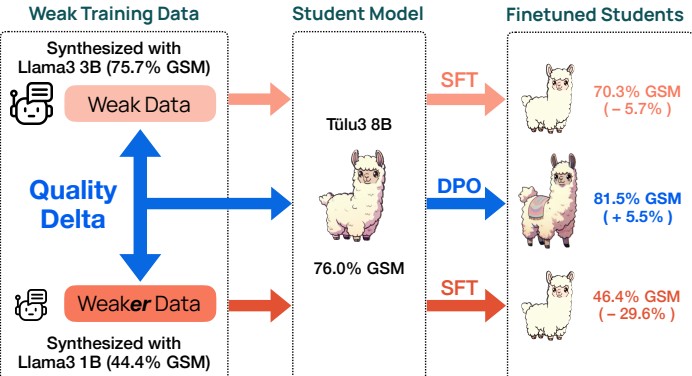

Figure 1: The **delta learning hypothesis** posits that paired preference data enables language models to learn from **relative differences** in data quality, driving gains beyond the absolute quality of each individual data point. Example: Tuning Tülu-3-8B-SFT to prefer greedy responses from Llama3 3B over those from Llama3 1B improves Tülu's GSM8k accuracy, even though both Llamas are weaker than Tülu on GSM8k. SFT on the weak data hurts.

## 1 Introduction

Common wisdom in machine learning holds that *strong data builds strong models*: improving performance typically requires training on data that exceeds a model's current capabilities. This principle has driven progress across the language model pipeline—from pretraining

corpus curation (Li et al., 2024; Penedo et al., 2024; OLMo et al., 2024), to rejection sampling for finetuning data (Dong et al., 2023; Adler et al., 2024), and to preference tuning, where human annotators identify the best model outputs as targets to tune towards (Ouyang et al., 2022; Bai et al., 2022). However, this wisdom also implies an inherent limitation: model capability may be upper-bounded by the strength of supervision available. Many desirable tasks are difficult to support with strong data, either because of high collection costs (e.g., synthesizing scientific literature at a PhD level) or because the task exceeds current human expertise (e.g., formulating a unified theory of physics). Thus, we ask: how might we build models that exceed the capabilities demonstrated in their training data?

In this paper, we show that preference pairs consisting of individually weak data points (e.g., responses from weak models) can be leveraged to improve a stronger language model *beyond* the strength of each individual sample. Our study is motivated by preliminary evidence in the literature (Yao et al., 2024; Zhu et al., 2024) and an intriguing pilot result: preference tuning a modern 8B Llama 3 (Dubey et al., 2024) large language model (LLM) using paired outputs from weaker, past-generation models consistently leads to performance gains, even when supervised finetuning on those same weak responses directly results in degradation.

We formalize these observations as the **delta learning hypothesis** (Figure 1), which posits that data with high *absolute* quality is not strictly necessary to improve language models. Instead, the relative quality difference—the "delta"—between paired samples can provide sufficient supervision to guide improvement through preference tuning, even if neither sample alone is stronger than the model being trained. Intuitively, the delta defines a meaningful direction of improvement; a strong model may learn to generalize along this direction and improve beyond the absolute quality of the preferred example. We systematically test our hypothesis in two controlled experiments by explicitly constructing preference pairs with limited absolute quality but a clear delta, and find consistent empirical evidence in support.

**Our hypothesis enables new open recipes for state-of-the-art language model post-training—without requiring any strong supervision**. To test the limits of "delta learning," we preference-tuned Tülu-3-8B-SFT, the instruction-finetuned precursor to Tülu 3, a state-of-the-art openly post-trained model (Lambert et al., 2024). In contrast to typical open recipes (Lambert et al., 2024; Ivison et al., 2023), which heavily distill from strong supervisors (e.g., GPT-4o) to generate high-quality chosen responses, we generate chosen responses with a single small model (e.g., Qwen 2.5 3B Instruct) that is not stronger than Tülu-3-8B-SFT itself. We pair these responses with outputs from an even small*er* model (e.g., Qwen 2.5 1.5B Instruct), thus creating a delta for learning. Strikingly, on a standard 11-benchmark evaluation suite, **our simple recipe matches Tülu 3's performance**, despite using vastly less supervision. Our analysis explains our recipe's success; we find that the chosen response only needs to meet a surprisingly low quality threshold (i.e., not significantly worse than base model), beyond which the size of the quality *delta* becomes the primary determinant of downstream preference tuning performance. Delta learning offers simple, cheap, and performant post-training, reducing the reliance of open recipes on strong model distillation.

To further illuminate *why* delta learning works, we theoretically study a logistic regression setup where a student model is trained to prefer synthetic pseudo-labels from a (possibly weak) teacher model over those from an even weaker one. We prove that even when both teachers provide misleading supervision individually, the performance gap between them ensures that the *delta between* these signals is still directionally correct. Learning from this delta can improve an already-strong student with high probability.

Overall, our work shows that models can learn surprisingly well from weak data, provided the data is paired to expose informative deltas. We find these deltas are often readily obtainable—even simple heuristics like model size differences suffice to capture them. Thus, we are optimistic that weak, currently unused data may be revitalized into valuable supervision. Furthermore, curating pairs of weak data may offer a more scalable alternative to finetuning in settings where strong supervision is limited—for example, by generating targeted corruptions to existing data or collecting lightweight human edits of weak model outputs. Finally, curating paired data may potentially enable training of superhuman models with preference labels on human-level outputs (Burns et al., 2023; Bowman et al., 2022). We leave these directions for future work.

| Model / Training | MMLU | AE2 | Full Avg. |
|---|---|---|---|
| LLAMA-3.2-3B-INST. | 62.9 | 18.7 | 57.8 |
| + UF-WEAK SFT | 61.8 | 12.3 | 54.0 |
| + UF-WEAK DPO | **64.0** | **22.4** | **59.0** |
| LLAMA-3.1-8B-INST. | 71.8 | 24.9 | 63.9 |
| + UF-WEAK SFT | 65.7 | 8.9 | 56.1 |
| + UF-WEAK DPO | **72.0** | **26.3** | **64.5** |

Table 1: We tune Llama 3 Instruct models on the ULTRAFEEDBACK-WEAK preference dataset, generated by models weaker than Llama 3. Training with preference learning (DPO)—to prefer "weak responses" over "weak*er* responses"—yields gains, while SFT directly on the weak preferred responses hurts performance. Blue indicates gain over baseline, orange degradation.

## 2 A Warm-up Case Study

We begin our investigation with an intriguing empirical finding: training on paired preference data generated by weak models can improve a stronger model's performance, even when finetuning directly on the weak models' outputs hurts.

**Data.** We start with ULTRAFEEDBACK, a popular preference dataset (Cui et al., 2023) consisting of preference pairs $(x, y_c, y_r)$, where $y_c$ and $y_r$ are an LLM-generated chosen and rejected response (respectively) to a prompt $x$. We filter to explicitly exclude all responses from models that have an LMSYS Chatbot Arena ELO score near or above Llama-3.2-3B-Instruct. Hence, the chosen response $y_c$ now derives from a model weaker than the Llama 3 models, although it is still higher-quality than the rejected response $y_r$. We call the resulting filtered dataset ULTRAFEEDBACK-WEAK. See Appendix G.2 for details.

**Training and evaluation.** We finetune Llama-3.2-3B-Instruct and Llama-3.1-8B-Instruct (Touvron et al., 2023) on ULTRAFEEDBACK-WEAK in two ways. One, we (1) preference tune with the DPO algorithm (Rafailov et al., 2024) on the preference pairs $(x, y_c, y_r)$. We compare to (2) supervised finetuning (SFT) directly on the chosen responses $(x, y_c)$. We evaluate models on 8 standard benchmarks that measure knowledge recall, mathematical reasoning, instruction following, truthfulness, general reasoning, and coding. See Appendix G.2 for full details of benchmarks used, along with training and hyperparameter details.

**Results.** We show a representative subset of results in Table 1, deferring the rest to the Appendix (Table A2). SFT on the chosen responses significantly hurts performance—likely because the models are finetuned to imitate weaker outputs. Yet surprisingly, preference tuning with the *same* weak preference pairs improves overall performance across benchmarks. Thus, regardless of absolute data quality, there may exist valuable learning signal in the pairwise contrast between chosen and rejected responses, which preference tuning can leverage. We will now develop this intuition into our central hypothesis.

## 3 The Delta Learning Hypothesis

We hypothesize that training on paired responses $(x, y_c, y_r)$ enables learning from the relative quality difference—the delta—between $y_c$ and $y_r$. Even if both responses $y_c, y_r$ have low absolute quality compared to the model we aim to improve, as long as $y_c$ is better than $y_r$ along some informative axes, the model can learn from this delta and improve.

Formally, let $\mu(x, y)$ be the utility of a response $y$ to some prompt $x$. In practice, $\mu$ may represent human preference, or simply some arbitrary function we wish to optimize. Suppose we wish to improve a model $M$. The **delta learning hypothesis** posits that there exist natural preference pairs $(x, y_c, y_r)$, where $\mu(x, y_c) > \mu(x, y_r)$, such that two conditions hold:

1. **Low absolute utility**: The utility $\mu(x, y_c)$ of the chosen response $y_c$ is no higher than the current capability of model $M$, and therefore supervised finetuning on $(x, y_c)$ explicitly hurts the model or at best does not help.

2. **Extrapolated gain**: Preference tuning on the pair improves model $M$ *beyond* $\mu(x, y_c)$.

We now present experiments with language models in controlled settings where we explicitly manipulate $\mu$ and construct responses $y_c, y_r$ of varying utility to test the delta learning hypothesis; we find consistent evidence in support. Later in Section 6, we theoretically study delta learning in logistic regression to better understand *why* delta learning can work.

### 3.1 Controlled Experiment: Stylistic Delta in Number of Bold Sections

We start with a toy setting where we explicitly define $\mu(x, y)$ to be "the number of Markdown-denoted bold section headers in $y$" (e.g., **example header**), a measurable and

| Model/Algorithm | Chosen Res. | Rejected Res. | Section Δ | # Sections Generated |
|---|---|---|---|---|
| LLAMA-3.2-3B-INST. (Baseline) | — | — | — | 5.9 |
| + SFT | 9 sections | — | — | 24.6 (+ 18.7) |
| + SFT | 3 sections | — | — | 4.4 (− 1.5) |
| + SFT | 2 sections | — | — | 2.9 (− 3.0) |
| + DPO | 3 sections | 2 sections | +1 | 81.1 (+ 75.2) |
| + DPO | 2 sections | 3 sections | −1 | 1.1 (− 4.8) |
| + DPO | 3 sections | 3 sections | 0 | 6.1 (+ 0.2) |

Table 2: We train Llama-3.2-3B-Instruct with DPO on preference pairs with responses containing a varying number of bold sections, and compare to SFT on the chosen response directly. When responses contain fewer sections than the model's baseline (i.e., < 5.9 sections), SFT decreases the number of sections generated. In contrast, preference tuning can leverage the *delta* between responses, increasing the number of sections generated even when each response is individually suboptimal.

controllable metric. Our hypothesis predicts that if we tune $M$ on preference pairs $(x, y_c, y_r)$ where $y_c$ contains, say, 3 sections and $y_r$ contains 2, then $M$ should learn to produce more sections—even though 3 sections (the "better" response) is fewer than $M$'s current capability, and hence would hurt when used as SFT data. As shown below, this is indeed observed.

**Setup.** We build a dataset of prompts $x$ matched with responses $y_{k_1} \ldots y_{k_n}$ containing varying numbers $k_i$ of bolded sections (details in Appendix G.3). We then tune Llama-3.2-3B-Instruct with DPO on preference pairs $(x, y_{k_i}, y_{k_j})$ formed by selecting a response $y_{k_i}$ with more sections ($k_i > k_j$) as chosen. To isolate potential confounding effects associated with preference tuning, we also consider two control settings: (1) reversing the preference pairs ($k_i < k_j$) and (2) tuning with responses containing an equal number of sections ($k_i = k_j$). We compare against SFT on the chosen responses $y_{k_i}$. See Appendix G.3 for hyperparameter details. We evaluate by measuring the average number of bolded sections generated before and after training in response to a set of held-out test prompts.

**Results.** Results in Table 2 strongly support our hypothesis: SFT only helps when the training responses are higher quality than the model's baseline "capability" (i.e., response contains more sections than what the model generates), and otherwise hurts. In contrast, even when responses are individually weak according to our defined $\mu$, pairing them together with a positive delta massively boosts section generation, extrapolating beyond the number of sections contained in the chosen response (see Figure A1—the model learns to make nearly every single word a new section header!). Our negative controls (preference tuning with negative delta or zero delta) do not yield gains; the positive delta is thus critical.

### 3.2 Controlled Experiment: Semantic Delta from a Weaker Model

We test whether our hypothesis extends beyond a one-dimensional style feature to general semantic quality by tuning a model $M$ to prefer its own self-generated greedy responses (which, by construction, do not exceed its capability) over responses from a weaker (smaller) model. Please see Appendix D.3 for results and analysis that further support our hypothesis.

## 4 Post-training Language Models with Delta Learning

Having validated our hypothesis in controlled settings, we now test its applicability in a realistic, large-scale setting: 8B LLM post-training. Current open recipes extensively rely on strong LLMs (e.g., GPT-4o) to generate preference data with high-quality chosen responses to learn from (Lambert et al., 2024; Ivison et al., 2023). However, the delta learning hypothesis suggests that preference tuning can be effective even when chosen responses are *not* high quality, provided we can construct a meaningful delta to a weaker rejected response. Pushing this idea to its logical extreme, we propose a simple preference tuning setup that explicitly eliminates the use of any strong LLMs (i.e., larger than 3B parameters) for either response generation or preference annotation from an existing state-of-the-art synthetic preference data recipe. Surprisingly, we find our simplifying changes incur little

performance trade-off, enabling a significantly cheaper and more accessible post-training recipe that reduces the reliance of open recipes on strong model distillation.

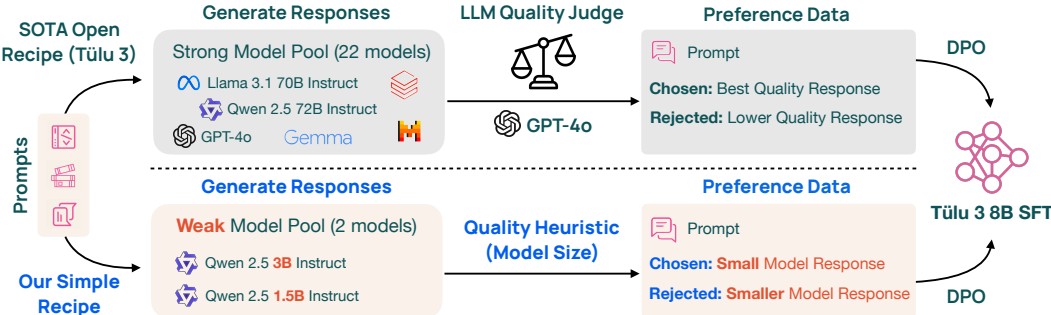

Figure 2: We simplify the **Tülu 3 preference data recipe** (top half). Instead of using a GPT-4o judge to pick the best response from many strong models as chosen, **our recipe (bottom half)** uses a single small model to generate all chosen responses, relying on the implicit delta to an even smaller (and thus weaker) model's responses to drive downstream learning.

## 4.1 A State-of-the-Art Existing Setup: The Tülu 3 Recipe

To contextualize our simplifications, we first detail Tülu 3 (Lambert et al., 2024), the current state-of-the-art recipe in open-source post-training. Tülu 3 comprises a series of 8B and 70B language models post-trained on top of the Llama 3 base models, achieving performance that matches or exceeds equivalently-sized proprietary models. Hence, we adopt the Tülu 3 8B preference tuning recipe (Figure 2, top half) as an ideal starting point for our exploration.

Tülu 3 preference data is constructed starting from 271k diverse prompts. Responses are generated using strong modern LLMs (e.g., Llama-3.1-70B-Instruct, Qwen-2.5-72B-Instruct (Yang et al., 2024a), etc.). A frontier LLM (GPT-4o) then scores these responses; preference pairs are formed by selecting the highest-scoring response as chosen and a lower-scoring one as rejected. This data is then used to DPO tune an intermediate instruction-finetuned model (Tülu-3-8B-SFT), yielding the preference tuned model Tülu-3-8B-DPO.

Besides the substantial cost of GPT-4o annotation (~$10,000 USD), the Tülu 3 preference tuning recipe fundamentally assumes access to supervision sources stronger than the model being trained (i.e., an 8B model), both for (1) generating high-quality responses (i.e., using 70B models) and (2) annotating response quality (using GPT-4o). As we demonstrate below, this assumption can be entirely eliminated.

## 4.2 Our Simple Recipe: Constructing Preference Data without Strong Supervision

Our recipe, illustrated in Figure 2 (bottom half), simplifies the Tülu 3 preference tuning recipe while keeping the starting model checkpoint (Tülu-3-8B-SFT) and initial prompts fixed to isolate our changes. We intervene by removing all use of strong models:

**Chosen response generation.** Starting from the same set of prompts as the Tülu 3 dataset, we generate all chosen responses with a single small model (e.g., Qwen 2.5 3B Instruct) that is near or below the capability of Tülu-3-8B-SFT (as measured by downstream evaluations, see below). With a 3B chosen model, this change reduces the FLOPs needed for data generation by over an order of magnitude (~6% of the original).

**Forming preference pairs.** We eliminate GPT-4o quality annotations entirely. Drawing from our findings in Section 3.2, we simply use model size as a proxy for quality. We pair every chosen response with a response from the next-smallest model in the same model family (e.g., pair Qwen 2.5 3B Instruct with Qwen 2.5 1.5B Instruct). While this heuristic is noisy—the smaller model might occasionally generate better responses—our previous controlled experiments show that learning can still occur with such noisy semantic deltas.

With our simplified pipeline, we construct three preference datasets, generating chosen responses with either (1) Qwen-2.5-3B-Instruct (paired with Qwen-2.5-1.5B-Instruct), (2)

| Model/Preference Data | MMLU | PopQA | MATH | GSM | AE2 | IFEval | BBH | DROP | TQA | HEval | HEval+ | Avg. |
|---|---|---|---|---|---|---|---|---|---|---|---|---|
| LLAMA-3.2-1B-INSTRUCT | 46.1 | 13.9 | 21.1 | 44.4 | 8.8 | 54.5 | 40.2 | 32.2 | 40.0 | 64.8 | 60.0 | 38.7 |
| LLAMA-3.2-3B-INSTRUCT | 62.9 | 19.4 | 39.6 | 75.7 | 18.7 | 76.5 | 61.6 | 48.5 | 50.6 | 79.7 | 76.8 | 55.5 |
| QWEN-2.5-0.5B-INSTRUCT | 46.2 | 10.1 | 27.2 | 39.2 | 3.3 | 28.8 | 32.2 | 25.3 | 45.4 | 60.5 | 58.9 | 34.3 |
| QWEN-2.5-1.5B-INSTRUCT | 59.7 | 15.4 | 41.6 | 66.2 | 7.2 | 44.2 | 45.9 | 14.1 | 46.5 | 83.0 | 79.8 | 45.8 |
| QWEN-2.5-3B-INSTRUCT | 69.5 | 15.7 | 63.1 | 77.7 | 17.8 | 64.0 | 57.6 | 31.5 | 57.2 | 90.5 | 87.4 | 57.5 |
| TÜLU-3-8B-SFT | 66.1 | 29.6 | 31.2 | 76.0 | 12.2 | 71.3 | 69.2 | 61.2 | 46.8 | **86.2** | 79.8 | 57.2 |
| + Llama 3.2 3B over 1B | 68.8 | 30.3 | 40.9 | 81.5 | 24.9 | 75.0 | 70.0 | 60.7 | 54.2 | 84.7 | 81.1 | 61.1 |
| + Qwen 2.5 1.5B over 0.5B | 67.4 | 29.9 | 39.9 | 79.8 | 15.8 | 72.5 | **70.8** | 61.8 | 52.1 | 83.7 | 78.1 | 59.3 |
| + Qwen 2.5 3B over 1.5B | 69.4 | **31.7** | 42.6 | 83.4 | **36.1** | 78.6 | 69.4 | 62.0 | **57.7** | 84.4 | **81.7** | **63.4** |
| + Tülu 3 Preference Dataset | **69.8** | 30.3 | 42.6 | **84.2** | 32.8 | **80.4** | 69.2 | **62.5** | 56.1 | 84.7 | 80.8 | 63.0 |

Table 3: We train Tülu-3-8B-SFT with DPO on preference data constructed with our simple recipe, which pairs outputs from a weak model (chosen response) with outputs from an even weak*er* model (rejected). Strikingly, our best setup matches the original Tülu 3 preference data, which requires vastly stronger supervision (e.g., from GPT-4o). We generate our data using models that are near or below Tülu-3-8B-SFT in average performance (top half).

Qwen-2.5-1.5B-Instruct (paired with 0.5B), or (3) Llama-3.2-3B-Instruct (paired with Llama-3.2-1B-Instruct). We choose these exact models because the original pool of data-generating models in Tülu 3's preference data recipe explicitly includes larger models from both the Qwen 2.5 and Llama 3 model families, while excluding these small models. Hence, the chosen responses in the original Tülu 3 preference data are of significantly higher absolute quality compared to our setup (i.e., 4.44/5 absolute quality points as judged by GPT-4o, versus 3.98/5 in our setup with Qwen-2.5-3B-Instruct generating chosen responses). Our setup thus relies more on the **delta** between chosen and rejected responses to drive learning. See Appendix E for qualitative examples of these deltas, as well as additional statistics of our datasets (e.g., response length, vocabulary diversity, etc.).

### 4.3 Matching Tülu with Delta Learning

We preference tune Tülu-3-8B-SFT on our weak preference datasets using DPO, tuning hyperparameters following Lambert et al. (2024) (see Appendix G.5). We evaluate our models on all benchmarks from Section 2 as well as three additional benchmarks measuring model capabilities for consistency with Tülu 3's evaluations (see Appendix B). We show further results on six safety evaluations in Appendix C. To compare our simple preference data against the Tülu 3 preference data, we evaluate the official Tülu-3-8B-DPO model.

Our main results in Table 3 reveal a striking finding: **tuning with our simple weak preference data recipe matches the original Tülu 3 recipe in performance**, achieving a +0.4 point average gain over the Tülu 3 preference data when using Qwen-2.5-3B-Instruct to generate chosen responses. Even though Tülu 3 preference data uses strong model supervision to synthesize chosen responses of significantly higher absolute quality, the quality delta between chosen and rejected responses in our weak pairs still suffices to produce comparable gains—it is possible to learn a surprising amount from the quality delta alone.

Consistent with our hypothesis, we observe gains when tuning with any of our three weak datasets. For instance, tuning with chosen responses from Qwen-2.5-1.5B-Instruct—which is 11.4 points worse than Tülu-3-8B-SFT on average—still yields a +2.1 point gain in average performance. The delta learning phenomenon also holds on the level of individual tasks; for example, tuning on Llama-3.2-3B-Instruct chosen responses boosts GSM8K accuracy by 5.5 points, despite Llama being weaker than Tülu on GSM8K. Finally, our preference data recipe yields gains when using either Llama or Qwen models to generate preference data, suggesting that it is not reliant on idiosyncrasies of a specific model family.

## 5 Analysis

We identify four factors in our simple recipe that may impact preference tuning performance and study each: (1) the magnitude of the quality delta between chosen and rejected responses, (2) the absolute quality of chosen responses, (3) our model size-based reward heuristic, and (4) our choice of Tülu-3-8B-SFT as the base model to tune from. We use the

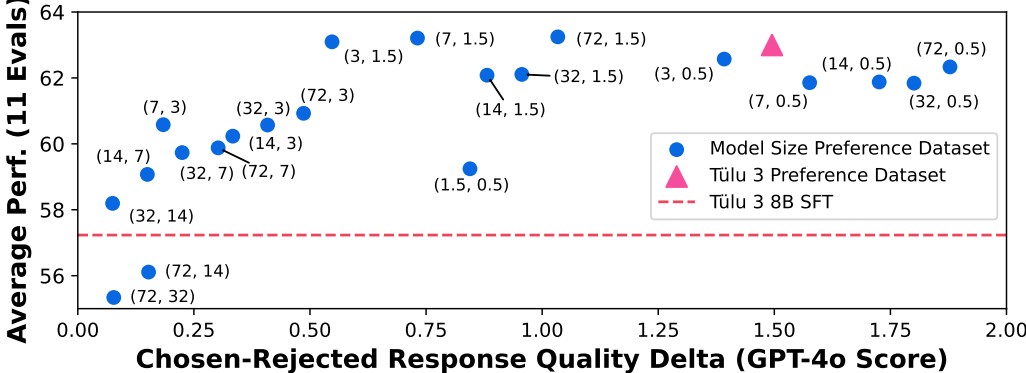

Figure 3: The size of the quality delta between chosen and rejected responses is a strong predictor of downstream preference tuning performance. Performance improves as the delta increases, up to approximately $\Delta \approx 0.55$, beyond which gains plateau. Each dot in the plot represents a model preference-tuned from Tülu-3-8B-SFT, using either the **original Tülu 3 data** or data generated by pairing two Qwen 2.5 Instruct models following **our simple recipe**. Numbers in parentheses indicate the parameter counts (B) of the paired models.

same 11-benchmark evaluation as in Section 4. Unless noted otherwise, all models are tuned from Tülu-3-8B-SFT with DPO. We defer full training details to Appendix G.6.

### 5.1 How does the magnitude of the chosen-rejected quality delta affect learning?

Using our simple recipe, we construct 21 preference tuning datasets containing chosen and rejected responses with varying absolute quality (and thus varying deltas). Starting with Tülu 3 prompts, we synthesize preference pairs with all 21 possible model pairs from the Qwen 2.5 Instruct family, which we select for its wide variety of model sizes (0.5B, 1.5B, 3B, 7B, 14B, 32B, 72B). Following our recipe, we select the larger model's response as chosen (e.g., 72B over 7B). To quantify the absolute quality of these responses, we apply the GPT-4o annotation method from Lambert et al. (2024) to score 10k responses from each model on a 1–5 scale. We plot average performance after preference tuning against the average pairwise delta between chosen and rejected responses in Figure 3. Our results suggest that:

**The magnitude of the delta strongly predicts downstream preference tuning performance, with a minimal delta required for optimal gains.** We observe a strong positive correlation between the magnitude of the delta and downstream performance, up until approximately $\Delta \approx 0.55$, after which performance peaks and plateaus. Beyond this threshold, further increases in the delta do not yield additional downstream gains. Interestingly, **the Tülu 3 preference dataset falls in line with this same trend**, despite being generated via a significantly different recipe. This alignment provides some evidence for why our simple recipe can match Tülu 3 in performance: both the Tülu 3 dataset and our Qwen-2.5-3B-Instruct-generated weak data exhibit deltas that are above the saturation threshold, after which nearly all datasets perform comparably.

An outlier to this trend is the dataset with Qwen-2.5-1.5B-Instruct-generated chosen responses. The gain over Tülu-3-8B-SFT from tuning on this data is positive, but smaller than what its delta size alone would predict. We conjecture that this is because Qwen 1.5B is the only chosen model considered that is *substantially* weaker than the Tülu-3-8B-SFT base.

**Not all positive deltas drive learning.** Tuning on preference datasets generated by pairing the 72B Qwen model with 32B or 14B-sized models *hurts* performance—these are the only two points that fall below the Tülu-3-8B-SFT dashed line in Figure 3. We observe that the log-likelihoods of both the chosen and rejected responses decrease during DPO training on these datasets (an occasional phenomenon in DPO training, see Razin et al. (2024)). We hypothesize that this effect may be particularly harmful when both chosen and rejected responses are much stronger than the base model being tuned; the model is optimized to downweight good behaviors. It remains an open question as to what properties of the

| Model/Preference Data | MMLU | PopQA | MATH | GSM | AE2 | IFEval | BBH | DROP | TQA | HEval | HEval+ | Avg. |
|---|---|---|---|---|---|---|---|---|---|---|---|---|
| TÜLU-3-8B-SFT | 66.1 | 29.6 | 31.2 | 76.0 | 12.2 | 71.3 | 69.2 | 61.2 | 46.8 | **86.2** | 79.8 | 57.2 |
| + Model size heuristic | 69.4 | **31.7** | **42.6** | 83.4 | **36.1** | 78.6 | **69.4** | **62.0** | 57.7 | 84.4 | **81.7** | 63.4 |
| + GPT-4o judge reward | **69.9** | 31.5 | 40.6 | 83.9 | 29.9 | **79.5** | 66.5 | 61.2 | **62.4** | 85.7 | 80.8 | 62.9 |
| OLMO-2-7B-SFT | 61.4 | **23.6** | 25.3 | 73.5 | 8.4 | 66.5 | 49.3 | 59.6 | 48.6 | 70.0 | 63.8 | 50.0 |
| + Qwen 2.5 3B over 1.5B | **62.9** | **23.6** | 30.0 | 80.6 | **31.0** | 71.5 | 50.9 | 59.3 | 56.3 | **72.6** | **66.6** | 55.0 |
| + OLMo 2 Preference Dataset | 61.9 | 23.5 | **30.3** | **83.1** | 27.7 | **72.3** | 50.9 | **60.2** | 56.0 | 70.7 | 66.2 | 54.8 |

Table 4: **Top half.** We ablate our use of model size as a heuristic to label preference pairs in our simple recipe, and find that it is a good proxy for GPT-4o judge reward. **Bottom half.** We ablate our choice of tuning from Tülu-3-8B-SFT. We find that our setup generalizes to OLMo-2-7B-SFT, matching the original OLMo 2 preference recipe's performance.

delta—beyond magnitude—are necessary to drive effective learning, and how these factors interact with the choice of preference tuning algorithm.

## 5.2 How does the chosen response's absolute quality affect learning?

We group the 21 preference datasets described above (Section 5.1) according to the strength of the Qwen 2.5 model used to generate the chosen response (i.e., 1.5B, 3B, 7B, 14B, 32B, or 72B parameters). For each group, we selected the dataset that yields the highest downstream performance after preference tuning as a best-case measure for the performance achievable when tuning on chosen responses generated by a model of a given strength. For comparison, we also evaluated supervised finetuning directly on the responses generated from each of the Qwen models. We defer results and full analysis to Appendix D.4. In summary, we find that SFT only yields gains when tuning on responses from Qwen models stronger than our base Tülu-3-8B-SFT model, with gains scaling monotonically as the data-generating model's strength increases. In contrast, preference tuning on chosen responses of any quality yielded strong gains—even when generated by models weaker than or equal to our base model.

## 5.3 Ablations

We ablate our choice of (1) **using model size as a preference heuristic** (as opposed to annotating preferences with an LLM judge) and (2) **tuning from Tülu-3-8B-SFT**. We report results in Table 4 and highlight key takeaways, deferring full experimental details and discussion to Appendix D.5. First, our simple recipe generalizes well to tuning OLMo-2-7B-SFT, matching the performance of the Olmo 2 Preference Data (constructed similarly to the Tülu 3 data). Second, **the model size heuristic is a surprisingly accurate proxy for GPT-4o preferences**, with an 80.5% agreement rate. Finally, learning from the delta between weak responses can succeed independently of the specific preference signal used. See Appendix D.6 for a further ablation on our choice of preference tuning algorithm; we find that delta learning also succeeds with SimPO (Meng et al., 2024).

## 6 Delta Learning in Logistic Regression, Provably

We have shown empirical evidence of our hypothesis (Section 3) and utilized it for performant language model post-training (Section 4). Now, we study *why* delta learning works. We study intercept-free binary logistic regression where a student is preference-tuned to prefer pseudo-labels from one teacher over pseudo-labels from a *weaker* one. We prove that this teacher strength gap alone can guarantee that the student improves with high probability, even when both teachers are weaker than the student.

### 6.1 Problem Setup and Notation

**Preliminaries.** Suppose inputs are drawn as $x \sim \mathcal{N}(0, I_d)$ and that labels $y^* \in \{0, 1\}$ are assigned via some ground-truth unit-norm parameter $\theta^* \in \mathbb{R}^d$, $y^* = \mathbb{1}\{\langle \theta^*, x \rangle \geq 0\}$. Because the input distribution is an isotropic Gaussian, the population 0-1 loss incurred by any model $\theta \in \mathbb{R}^d$ depends only on its angle with $\theta^*$: $\mathcal{L}_{\text{0-1}}(\theta) := \Pr_{x \sim \mathcal{N}(0, I_d)} \left[ \text{sgn} \langle \theta, x \rangle \neq \text{sgn} \langle \theta^*, x \rangle \right] =$

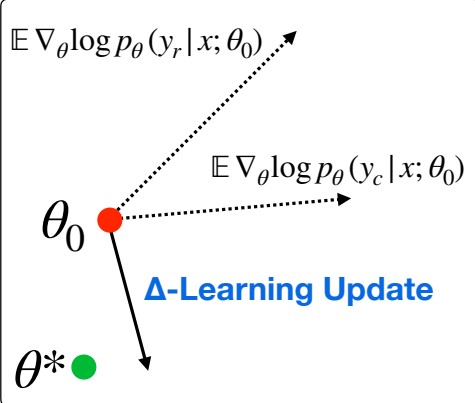

$\mathbb{E}\nabla_\theta \log p_\theta(y_r|x;\theta_0)$

$\mathbb{E}\nabla_\theta \log p_\theta(y_c|x;\theta_0)$

$\theta_0$

**Δ-Learning Update**

$\theta^*$

Figure 4: We train **student model** $\theta_0$ to prefer pseudo-labels $y_c$ from a (possibly weak) teacher model $\theta_c$ over pseudo-labels $y_r$ from a *weaker* teacher $\theta_r$. Individually, the gradient updates $\nabla_\theta \log p_\theta(y_c|x;\theta_0), \nabla_\theta \log p_\theta(y_c|x;\theta_0)$ (dashed arrows) induced by the weak labels can be harmful in expectation and steer the student away from the **ground-truth parameters** $\theta^*$. In contrast, the **delta learning gradient updates** follow their *difference* vector $\nabla_\theta \log p_\theta(y_c|x;\theta_0) - \nabla_\theta \log p_\theta(y_r|x;\theta_0)$, which is positively aligned with $\theta^*$ whenever $\theta_c$ outperforms $\theta_r$. Pairing weak-and-weaker teachers thus yields a learning signal that can improve a stronger student.

$\frac{1}{\pi}\arccos\frac{\langle\theta,\theta^*\rangle}{\|\theta\|_2}$. Thus, the classification accuracy—or *strength*—of any model $\theta$ is proportional to its cosine similarity with $\theta^*$, denoted $\cos(\theta,\theta^*)$.

**Delta learning training.** Fix a student model $\theta_0 \in \mathbb{R}^d$, and fix two teachers $\theta_c, \theta_r$. Write $\alpha_0, \alpha_c, \alpha_r$ to denote the cosine similarity of each model with $\theta^*$, respectively. Assume that $\alpha_c > \alpha_r$, so that teacher $\theta_c$ is stronger than $\theta_r$ in expectation over the data. We have no assumption on the strength $\alpha_0$ of student $\theta_0$. For any input $x$, we assign chosen and rejected *pseudo-labels* as $y_c = \mathbb{1}\{\langle\theta_c, x\rangle \geq 0\}, y_r = \mathbb{1}\{\langle\theta_r, x\rangle \geq 0\}$, forming a preference pair $(x, y_c, y_r)$ annotated as $y_c \succ y_r$. Given a batch of $B$ preference pairs $\{(x^{(i)}, y_c^{(i)}, y_r^{(i)})\}_{i=1}^B$ and $p_\theta(y = 1|x) = \sigma(\langle\theta, x\rangle)$, we update student $\theta_0$ with SGD on a naïve preference loss: $\theta_{t+1} \leftarrow \theta_t - \eta \sum_{i=1}^m \nabla_\theta \mathcal{L}_{\text{pref}}(x^{(i)}, y_c^{(i)}, y_r^{(i)}; \theta_t)$, $\mathcal{L}_{\text{pref}}(x, y_c, y_r; \theta) := -(\log p_\theta(y_c|x) - \log p_\theta(y_r|x))$. The loss $\mathcal{L}_{\text{pref}}$ can be seen as an unnormalized SimPO-style loss (Meng et al., 2024); we drop the normalization for theoretical simplicity.

## 6.2 Delta Learning Succeeds with High Probability

Our central claim is that in sufficiently high dimensions, delta learning in logistic regression works with high probability. At a high level, we show that given any student model $\theta_0$ and under mild conditions, most pairs of teacher models $\theta_c, \theta_r$ exhibiting a performance delta (i.e., $\theta_c$ has higher accuracy than $\theta_r$) suffice to generate preference data that will improve the student model, even *beyond* the performance of each teacher.

Intuitively, preference tuning pushes the student's parameters towards the stronger teacher $\theta_c$ and away from the weaker teacher $\theta_r$. Since $\theta_c$ is (by assumption) better aligned with the ideal parameters $\theta^*$ than $\theta_r$, the difference vector $\theta_c - \theta_r$ is itself positively aligned with $\theta^*$, regardless of how low the absolute alignment of $\theta_c$ may be. In other words, **the delta between the two teachers yields a directionally correct signal, even when both teachers are individually weak** (Figure 4). As long as this useful signal is not swamped out by other spurious signals arising from the teachers' errors, tuning will improve the student. In particular, the spurious signals are most problematic when they align with and amplify the student's existing errors. Fortunately, the teachers' errors are essentially orthogonal to the student's errors in high dimensions, so such amplification rarely happens. This high-dimension effect creates a training length "sweet spot," where the student can improve from the useful signal without overfitting to the teachers' errors. We formalize this below.

**Theorem 6.1** (Delta Learning for Logistic Regression). *Fix a failure probability $\delta \in (0, 1)$, and let $\theta_0 \in \mathbb{R}^d$ be an arbitrary student model whose initial cosine similarity with the (unobserved) ground-truth model $\theta^*$ is $\alpha_0 := \cos(\theta_0, \theta^*) < 1, \theta_0 \neq \theta^*$. We train $\theta_0$ with delta learning following the setup of Section 6.1: given two teacher models $\theta_c, \theta_r$ satisfying*

$$\cos(\theta_c, \theta^*) =: \alpha_c, \quad \cos(\theta_r, \theta^*) =: \alpha_r, \qquad \alpha_c > \alpha_r,$$

we update $\theta_0$ to prefer teacher $\theta_c$'s pseudo-labels over the weaker teacher $\theta_r$'s via SGD on the naïve preference loss $\mathcal{L}_{pref}$. Then if the teachers and student satisfy

$$\kappa := \underbrace{(\alpha_c - \alpha_r)(1 - \alpha_0^2)}_{\text{useful signal from delta}} - \alpha_0 \underbrace{\langle \mathrm{Proj}_{(\theta^{*\perp})}(\tilde{\theta}_0), \mathrm{Proj}_{(\theta^{*\perp})}(v_\Delta) \rangle}_{\text{spurious noise orthogonal to } \theta^*} > 0, \tag{C1}$$

$$v_\Delta := (\theta_c / \|\theta_c\|_2) - (\theta_r / \|\theta_r\|_2), \qquad \tilde{\theta}_0 := \theta_0 / \|\theta_0\|_2, \tag{1}$$

and the ambient dimension exceeds a threshold of $d \gtrsim \ln\left[\left(\kappa + \|v_\Delta\|_2^2\right) / \left(\delta^2 \kappa \|v_\Delta\|_2^2\right)\right]$, training for $T$ total steps with batch size $B = \Theta(d)$ and learning rate $\eta$ where

$$\eta = \widetilde{\Theta}\left(\kappa^2 \|\theta_0\|_2 \cdot \min\left\{1/\sqrt{d}, \kappa / \|v_\Delta\|_2^2\right\}\right), \qquad T = (\kappa \|\theta\|_2) / \left(4\eta \|v_\Delta\|_2^2\right), \tag{2}$$

yields (with probability at least $1 - \delta$) a student iterate $\theta_T$ satisfying

$$\cos(\theta_T, \theta^*) > \cos(\theta_0, \theta^*) + \Theta(\kappa^2). \tag{3}$$

Hence, the trained model $\theta_T$ incurs strictly smaller population 0-1 loss than the initial student $\theta_0$.

Note that the right-hand side of Condition C1 can be made small regardless of the teachers' performance level; learning can succeed even when the initial student already outperforms both teachers, $\alpha_0 > \alpha_c > \alpha_r$. In fact, most teacher pairs satisfy Condition C1 in high dimensions, yielding our main result:

**Corollary 6.2** (Informal). *In high dimensions, most pairs of teacher models with a performance gap suffice to improve student $\theta_0$ via delta learning. Randomly sample two teacher models $\theta_c$, $\theta_r$ uniformly from all unit vectors, conditional on their cosine similarities $\alpha_c$, $\alpha_r$ with $\theta^*$, $\alpha_c > \alpha_r$. Then so long as the dimension $d$ exceeds a mild threshold, with high probability Condition C1 holds and by Theorem 6.1 training strictly improves $\theta_0$ with high probability.*

We defer all formal proofs and further discussion to Appendix F.

## 7 Related Work

**Learning from preference feedback.** Early preference tuning used reinforcement learning from human feedback (Ziegler et al., 2019; Ouyang et al., 2022; Bai et al., 2022), which involves training a reward model on human-annotated rankings of model outputs that is then optimized against with algorithms like PPO (Schulman et al., 2017). Recent work has simplified this approach by (1) removing the reward model in favor of direct policy updates (Rafailov et al., 2024; Meng et al., 2024; Ethayarajh et al., 2024) and (2) replacing human annotations with strong LLM judges (Cui et al., 2023; Lee et al., 2023). While the source of supervision has evolved, its use remains largely unchanged: modern approaches (Dubey et al., 2024; Lambert et al., 2024) still often rely on strong judges to select high-quality responses. We show that learning can succeed even when the chosen response is weak.

**Weak-to-strong generalization.** As models advance, the field is exploring ways to supervise them beyond human capability (Burns et al., 2023). Prior work has focused on eliciting behavior from base models (Hase et al., 2024; Burns et al., 2023) or enabling models to iteratively improve their own training data (Yang et al., 2024b; Wu et al., 2024). We show an alternative approach: leveraging relative differences in weak data to guide generalization.

## 8 Conclusion

In this work, we have shown that models can learn surprisingly well from the delta between paired weak data points. We further characterized key factors of paired data that drive learning, such as the delta's magnitude and the chosen response's absolute quality. We find that not all deltas are equally useful: some fail to drive gains, and gains saturate as chosen response quality improves. Natural questions arise: what makes a delta informative? How can we effectively scale delta-based learning? And to what extent are these dynamics dependent on the specific task or tuning algorithm? We leave exploration to future work.

# 9 Reproducibility

To ensure clean reproducibility, we provide extensive details on our training codebase and setup (Appendix G.1), our evaluation codebase and setup (Appendix B), our compute usage (Appendix H), and experiment-specific details for all experiments in our study (Appendix G), such as training hyperparameters and details in dataset creation. We encourage the reader to review the referenced sections.

## Acknowledgements

We thank Marco Tulio Ribeiro for for insights on an earlier version of the work. We further thank (alphabetically) Victoria Graf, Stella Li, Rulin Shao, Rui Xin, and Zhiyuan Zeng for useful discussions and proofreading. We further thank Stella and Victoria for their artistic expertise in iterating on our visualizations. We thank Ananya Harsh Jha for solidarity. SG is supported by the NSF GRFP. This work was supported by the Singapore National Research Foundation and the National AI Group in the Singapore Ministry of Digital Development and Information under the AI Visiting Professorship Programme (award number AIVP-2024-001), the AI2050 program at Schmidt Sciences, and the Google ML and Systems Junior Faculty Award.

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

## A  Limitations

As an empirical study with limited compute, our results are largely based on in-depth analysis over a single preference tuning algorithm (DPO) and a few base models (Llama 3, Tülu 3, OLMo 2). Moreover, our evaluation does not capture all potential downstream model behaviors. For example, we do not evaluate multilingual capabilities or domain-specific use cases such as scientific writing. As such, extending our findings to (1) other preference tuning algorithms, (2) larger model scales and different base models, and (3) new tasks are all interesting directions for future work.

## B  Evaluation Details

Unless otherwise noted, we evaluate all models on the following core set of 8 standard benchmarks. We provide the skill that each benchmark measures as well as abbreviations used in parenthesis. Following Tülu 3 (Lambert et al., 2024), we evaluate with the OLMES (Gu et al., 2024) implementation of these benchmarks, with the exact same evaluation configurations (e.g., for prompts, metrics, few-shot examples, etc.) for all benchmarks. We defer readers to the above references for further details.

- **MMLU (knowledge recall)** (Hendrycks et al., 2020)
- **MATH (mathematical reasoning)** (Hendrycks et al., 2021)
- **GSM8k (GSM; mathematical reasoning)** (Cobbe et al., 2021)
- **IFEval (instruction following)** (Zhou et al., 2023)
- **AlpacaEval 2 (AEval2, AE2; instruction following)** (Dubois et al., 2024)
- **TruthfulQA (TruthQA, TQA; truthfulness)** (Lin et al., 2021)
- **BigBenchHard (BBH; general reasoning)** (Suzgun et al., 2022)
- **Codex HumanEval+ (HEval+; coding)** (Liu et al., 2023)

For our post-training experiments (Section 4 and Section 5), we extend our evaluation suite to include three additional benchmarks to maintain evaluation consistency with the full suite from (Lambert et al., 2024). The added benchmarks are:

- **PopQA (knowledge recall)** (Mallen et al., 2022)
- **DROP (general reasoning)** (Dua et al., 2019)
- **Codex HumanEval (HEval; coding)** (Chen et al., 2021)

| Model / Preference Data | Unsafe Prompt Refusal | | | Jailbreak Resistance | | | Aggregate Scores | | |
|---|---|---|---|---|---|---|---|---|---|
| | XSTest | HarmB. | WildGuard | DAN | JailTrig. | WildJail. | Avg. Refusal | Avg. Jailbreak | Avg. All |
| LLAMA-3.2-1B-INSTRUCT | 81.1 | 65.0 | 78.1 | 87.0 | 74.5 | 61.8 | 74.7 | 74.4 | 74.6 |
| LLAMA-3.2-3B-INSTRUCT | 90.9 | 77.8 | 88.1 | 95.0 | 78.0 | 68.4 | 85.6 | 80.5 | 83.0 |
| QWEN-2.5-0.5B-INSTRUCT | 72.2 | 70.3 | 72.5 | 64.7 | 84.8 | 50.9 | 71.7 | 66.8 | 69.2 |
| QWEN-2.5-1.5B-INSTRUCT | 71.8 | 94.7 | 79.8 | 77.7 | 82.0 | 53.6 | 82.1 | 71.1 | 76.6 |
| QWEN-2.5-3B-INSTRUCT | 87.8 | 91.2 | 83.2 | 49.0 | 67.8 | 56.0 | 87.4 | 57.6 | 72.5 |
| TÜLU-3-8B-SFT | 90.7 | **98.8** | 99.2 | **87.7** | **96.0** | 86.6 | 96.2 | **90.1** | **93.2** |
| + Llama 3B over 1B | 93.1 | 97.2 | 99.2 | 73.0 | 88.0 | 85.1 | **96.5** | 82.0 | 89.3 |
| + Qwen 1.5B over 0.5B | 90.9 | 98.1 | **99.3** | 84.7 | 94.8 | **88.0** | 96.1 | 89.2 | 92.6 |
| + Qwen 3B over 1.5B | **93.6** | 96.2 | 99.1 | 62.3 | 83.0 | 78.8 | 96.3 | 74.7 | 85.5 |
| + Tülu 3 Preference Data | 92.9 | 95.3 | 98.5 | 68.7 | 87.2 | 81.3 | 95.6 | 79.1 | 87.3 |

Table A1: We evaluate the safety of the models post-trained with our weak preference data from Tülu-3-8B-SFT (Section 4) on six benchmarks measuring (a) whether the models refuse unsafe requests or (b) whether the models are robust to jailbreaking prompts. On all benchmarks, a higher score is better. The safety performance of the models we used to generate data are shown in the top half of the table.

## C  Additional Safety Evaluations for Post-trained Models

Following (Lambert et al., 2024), we further evaluated the models from our main post-training experiments (Section 4) on six safety benchmarks measuring either (a) whether models refuse to respond to unsafe requests or (b) whether models are robust to jailbreaking prompts. We list the benchmarks below, with skill measured in parenthesis:

- **XSTest (refusal)** (Röttger et al., 2023)
- **HarmBench (refusal)** (Mazeika et al., 2024)
- **WildGuardTest (refusal)** (Han et al., 2024)
- **Do-Anything-Now (abbreviated DAN; jailbreaking resistance)** (Shen et al., 2024)
- **JailbreakTrigger (jailbreaking resistance)** (Sun et al., 2024)
- **WildJailbreakTest (jailbreaking resistance)** (Jiang et al., 2024)

We show results of evaluation in Table A1. Overall, both strongly-supervised Tülu 3 preference data and delta learning with our weak preference data tend to slightly hurt average safety compared to the base SFT model. Thus, we conjecture that these drops are due to characteristics of the Tülu 3 prompt distribution (Lambert et al., 2024) shared between these data rather than an inherent limitation of delta learning. Intuitively, we speculate that if the prompts do not expose useful deltas, then one cannot expect gains. Nonetheless, models trained with delta learning degrade *less*, and hence are generally more safe than Tülu-3-8B-DPO. An exception is the model trained with Qwen-2.5-3B-Instruct over 1.5B responses. While it correctly refuses more often than Tülu-3-8B-DPO, it is easier to jailbreak. We conjecture that this is because Qwen 3B itself is easier to jailbreak than Qwen 1.5B (Table A1; see also (Biderman et al., 2023; OLMo et al., 2024) which suggest that model size can sometimes inversely correlate with safety), and hence the delta between Qwen 3B and 1.5B is in a negative direction. How to curate prompts and deltas that effectively improve safety remains an exciting open question.

## D  Extended Results and Discussions

### D.1  Pilot Study on ULTRAFEEDBACK-WEAK

See Table A2. Results are consistent with our findings in Section 2.

### D.2  Controlled Experiment: Stylistic Delta in Number of Bold Sections

Figure A1 shows examples of model generations before and after DPO training.

| Model/Training | MMLU | MATH | GSM | AEval2 | IFEval | BBH | TruthQA | HEval+ | Avg. |
|---|---|---|---|---|---|---|---|---|---|
| LLAMA-3.2-3B-INST. | 62.9 | 39.6 | 75.7 | 18.7 | **76.5** | **61.6** | 50.6 | **76.8** | 57.8 |
| + UF-WEAK SFT | 61.8 | 34.3 | 73.2 | 12.3 | 68.0 | 60.7 | 46.5 | 75.4 | 54.0 |
| + UF-WEAK DPO | **64.0** | **42.2** | **76.4** | **22.4** | 76.2 | 61.3 | **53.6** | 76.0 | **59.0** |
| LLAMA-3.1-8B-INST. | 71.8 | 43.0 | 83.7 | 24.9 | 78.2 | **72.7** | 55.1 | **81.6** | 63.9 |
| + UF-WEAK SFT | 65.7 | 34.6 | 77.9 | 8.9 | 59.9 | 71.8 | 49.1 | 80.8 | 56.1 |
| + UF-WEAK DPO | **72.0** | **43.9** | **83.9** | **26.3** | **81.1** | 72.3 | **56.2** | 80.4 | **64.5** |

Table A2: We tune Llama 3 Instruct models on the ULTRAFEEDBACK-WEAK preference dataset, generated by models weaker than Llama 3. Training with preference learning (DPO) to prefer "weak responses" over "weak*er* responses" yields gains, while SFT directly on the weak preferred responses hurts. Blue indicates gain over base model, orange degradation.

### D.3 Controlled Experiment: Semantic Delta from a Weaker Model

To test whether our hypothesis extends beyond a one-dimensional style feature to general semantic quality, we study the delta between self-generated outputs and outputs from a weaker model. Specifically, suppose we wish to improve model $M$. Given a set of prompts $\{x\}$, we can use $M$ to greedy decode responses $y_M = M(x)$. By construction, the quality of these responses exactly match $M$'s capability, $\mu(x, y_M) = \mu(x, M(x))$. Consequently, we would not expect SFT on $(x, y_M)$ to improve $M$'s overall performance. In contrast, our hypothesis predicts that creating a quality delta by pairing self-generated responses $y_M$ with semantically *weaker* responses $y_m$ may provide sufficient signal for preference tuning to improve $M$. A simple way to obtain such weaker responses is to use a smaller (weaker) model $m$ from the same model family as $M$ and greedy decode $y_m = m(x)$.

While similar in spirit to our pilot experiment (Section 2), this setup guarantees by construction that $M$ never observes any single chosen response of higher quality than it can currently produce. Our experiment also studies whether we can learn from a *noisy* delta, as even though $\mu(x, M(x)) > \mu(x, m(x))$ on average, some responses $y_m$ from the smaller model may surpass corresponding responses $y_M$ from the larger model on individual prompts.

**Setup.** We randomly sample 50k prompts $x$ from the Tülu 3 SFT dataset (Lambert et al., 2024). We greedy decode chosen responses $y_M$ with Llama-3.1-8B-Instruct (the model we later train) and rejected responses $y_m$ with Llama-3.2-3B-Instruct. We tune Llama-3.1-8B-Instruct with DPO to prefer its own responses $y_M$ over outputs $y_m$ from its smaller 3B sibling. We compare to SFT on $y_M$. As a negative control, we also preference tune to prefer $y_m$ over $y_M$. See Appendix G.4 for training details. We use the same evaluations from Section 2.

**Results.** Results in Table A3 further support our hypothesis. SFT on self-generated greedy responses reduces average performance by 1.2 points, possibly due to overfitting on these outputs at the expense of broader ability. In contrast, pairing self-generated responses with weaker responses creates a positive delta that drives learning beyond the baseline model's performance. This approach yields small but consistent gains on nearly all benchmarks, with a 0.4-point gain on average. Our negative control—flipping the preference order of self-generated and weaker responses—eliminates these gains and worsens overall performance ($-0.7$ points). Thus, the improvement comes specifically from the positive delta created by pairing with weaker responses, rather than general effects of preference tuning.

### D.4 Analysis: How much does Chosen Response Quality Matter?

We group the 21 preference datasets described in Section 5.1 according to the strength of the Qwen 2.5 model used to generate the chosen response (i.e., 1.5B, 3B, 7B, 14B, 32B, or 72B parameters). For each group, we selected the dataset that yields the highest downstream performance after preference tuning as a best-case measure for the performance achievable when tuning on chosen responses generated by a model of a given strength. For comparison, we also evaluated supervised finetuning directly on the responses generated from each of the Qwen models. Results in Figure A2 show that preference tuning generally outperformed

| Model/Training Setup | MMLU | MATH | GSM | AEval2 | IFEval | BBH | TQA | HEval+ | Avg. |
|---|---|---|---|---|---|---|---|---|---|
| LLAMA-3.2-3B-INSTRUCT (Weaker) | 62.9 | 39.6 | 75.7 | 18.7 | 76.5 | 61.6 | 50.6 | 76.8 | 57.8 |
| LLAMA-3.1-8B-INSTRUCT (Baseline) | 71.8 | 43.0 | 83.7 | 24.9 | 78.2 | 72.7 | 55.1 | 81.6 | 63.9 |
| + SFT (self-generated responses) | **72.2** | 42.2 | 82.9 | 24.3 | 76.3 | 72.1 | 53.4 | 78.0 | 62.7 |
| + DPO (self-generated over weaker) | 72.0 | **43.5** | **84.2** | **25.7** | **80.0** | 71.4 | **55.6** | **82.2** | **64.3** |
| + DPO (weaker over self-generated) | 70.9 | 42.3 | 83.4 | 22.9 | 78.6 | 72.1 | 54.6 | 80.5 | 63.2 |

Table A3: We train Llama-3.1-8B-Instruct using greedy responses generated by itself and by its weaker sibling, Llama-3.2-3B-Instruct. SFT on self-generated responses—which, by definition, equal the model's current capability—does not yield gains. In contrast, training with DPO to prefer self-generated responses over weaker ones can exploit the delta between them and yield consistent improvement. Reversing the preference order hurts performance; the positive delta is critical, not generic effects of preference tuning.

SFT; preference tuning with 3B chosen responses yielded higher gains than SFT on even 72B responses[1]. Moreover, we observe distinct qualitative trends for each tuning approach:

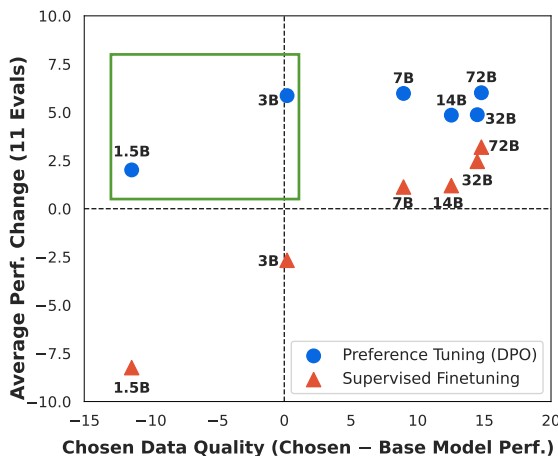

Figure A2: **Impact of chosen data quality**: We plot the performance of the model used to generate chosen responses (*x*-axis) against performance change after tuning on those responses (*y*-axis). **SFT** only yields gains when training on responses from models stronger than the base model we tune from; gains scale with chosen quality. **DPO** yields gains even when the chosen responses come from models no stronger than the base (**green box**); increasing chosen response quality gives quickly diminishing returns. Each dot represents a model tuned on chosen responses generated by a Qwen 2.5 Inst. model (parameter size labeled).

**Supervised finetuning: performance scales with chosen quality.** Performance gains (above *y*=0) only occur when tuning on responses from Qwen models stronger than our base Tülu-3-8B-SFT model (to the right of *x*=0). Gains scale monotonically with the chosen responses' absolute quality, increasing as the data-generating model's strength increases.

**Preference tuning: performance is less dependent on chosen quality, but saturates.** Tuning on chosen responses of any quality yielded gains—even when generated by models weaker than or equal to our base model (green box in figure). Still, chosen quality matters; tuning on chosen responses from Qwen 1.5B (much weaker than our base model) yields smaller gains than using stronger responses. However, once response quality reaches that of the tuned model (*x*=0), further improvements in chosen quality do not significantly improve performance. This saturation effect further explains why our weak preference data matches Tülu 3, despite using chosen responses with lower absolute quality.

### D.5 Extended Ablation Studies Analysis

**Model size preference heuristic.** We ablate our choice of using model size as a preference heuristic as opposed to annotating preferences with a LLM judge. Using Tülu 3's GPT-4o judge method, we re-labeled our best-performing weak preference dataset (responses from Qwen 2.5 3B Instruct paired with 1.5B responses). We find that **the model size heuristic is a surprisingly accurate proxy for GPT-4o preferences**. GPT-4o agrees with our model

---

[1]Tülu-3-8B-SFT has alredy been finetuned with even stronger supervision (e.g., human-written or GPT-4o responses), potentially diminishing the benefits of additional SFT.

size heuristic on 80.5% of the data; for comparison, GPT-4's agreement rate with human annotators has been estimated at around 65% (Dubois et al., 2023). Preference tuning with either GPT-4o preference labels or model size heuristic labels yielded comparable performance (Table 4, top half), except on AlpacaEval 2 (see discussion in next paragraph). Overall, our findings (1) validate our approach of using model size to ensure a chosen-rejected quality delta, and (2) show that learning from the delta between weak responses can succeed independently of the specific preference signal used.

**AlpacaEval 2 discrepancy.** The observed low performance on AlpacaEval 2 when using GPT-4o as a reward signal (Table 4) is likely because of the length-correction applied by the AlpacaEval 2 benchmark. LLM judges are known to have a bias towards preferring longer responses (Dubois et al., 2024); this bias is likely present when re-annotating our responses using the GPT-4o judge from Tülu 3. Hence, DPO training on preferences annotated by GPT-4o may increase the average generation length of the model, which is then penalized by the length-correction term that AlpacaEval 2 uses when computing winrate. Empirically, the model trained with GPT-4o preferences generates outputs that are around 200 characters longer on average compared to the model trained on the model size heuristic reward in response to the AlpacaEval 2 test prompts.

**Choice of Base Model.** To test the generality of our preference tuning recipe, we use it to tune a different base model, OLMo-2-7B-SFT. Using prompts from the OLMo 2 Preference Dataset—constructed in a nearly identical manner to the Tülu 3 data—we generate chosen and rejected responses using Qwen 2.5 3B Instruct and 1.5B, respectively. We choose this pair as it was the best configuration from Section 4. We tune OLMo-2-7B-SFT on both our simple data and the original OLMo 2 data. We show results in Table 4 (bottom half). Consistent with our earlier comparison against Tülu 3, our simple recipe matches the OLMo 2 preference data (+0.2 points average), which utilizes significantly stronger supervision.

### D.6 Additional Ablation Study: Preference Tuning Algorithm

We further ablate our choice of using DPO as the preference tuning algorithm in our main post-training experiments (Section 4) by instead tuning with SimPO (Meng et al., 2024) while keeping data and base model fixed. Specifically, we use SimPO to tune Tülu-3-8B-SFT on our best weak preference data (Qwen-2.5-3B-Instruct responses paired with 1.5B responses). We largely follow the same hyperparameters as our other analysis experiments (Appendix G.6); we additionally grid-sweep the following hyperparmaters:

- Dataset size: $\{100000, 200000\}$
- Learning rate: $\{5e\text{-}8, 7e\text{-}8, 1e\text{-}7, 3e\text{-}7\}$
- SimPO $(\beta, \gamma)$, roughly following the ranges tried in (Meng et al., 2024): $\{(10, 3.0), (5, 1.5), (2.5, 1.25), (2, 1.0)\}$

Results of training with SimPO are reported in Table A4. Overall, using SimPO to tune on weak data also yields strong gains, with a 5.2 point gain in average performance over the base SFT model. Consistent with (Lambert et al., 2024), the gains with SimPO are slightly less than with DPO (-1 point on average).

| Model/Preference Data | MMLU | PopQA | MATH | GSM | AE2 | IFEval | BBH | DROP | TQA | HEval | HEval+ | Avg. |
|---|---|---|---|---|---|---|---|---|---|---|---|---|
| TÜLU-3-8B-SFT | 66.1 | 29.6 | 31.2 | 76.0 | 12.2 | 71.3 | 69.2 | 61.2 | 46.8 | **86.2** | 79.8 | 57.2 |
| + DPO (Qwen 3B over 1.5B) | **69.4** | **31.7** | **42.6** | **83.4** | **36.1** | **78.6** | 69.4 | **62.0** | 57.7 | 84.4 | 81.7 | **63.4** |
| + SimPO (Qwen 3B over 1.5B) | 67.8 | 31.1 | 40.6 | 81.8 | 28.4 | 77.8 | **70.6** | 61.4 | **57.9** | 86.0 | **82.6** | 62.4 |

Table A4: **Top half.** We ablate our use of DPO as our preference tuning algorithm, and find that tuning with SimPO can also yield gains with weak preference data.

## E Weak Preference Dataset Details

Here, we provide generation details and statistics for the weak preference datasets used in our main post-training experiments (Section 4). To better illustrate what deltas exist between

the chosen and rejected responses, we also provide qualitative examples of preference pairs from our best performing weak dataset (Qwen-2.5-3B-Instruct over 1.5B responses).

## E.1 Dataset Creation Details

We find that the original Tülu 3 dataset contains approximately 6000 duplicated prompts (but not duplicated preference pairs, as Tülu 3 uses a large pool of models to generate responses and hence can form multiple distinct pairs for each prompt). Because our setup uses the same model to generate chosen responses for every prompt, we de-duplicated the repeated prompts, leaving 264806 remaining preference pairs for each of our weak datasets. These are the seed prompts are used for our simple recipe in Section 4.

## E.2 Qualitative Examples

To better understand the delta between weak and weaker responses, we manually inspected pairs from our best post-training preference data (i.e., chosen responses generated by Qwen-2.5-3B-Instruct and rejected responses by Qwen-2.5-1.5B-Instruct). Overall, there were no universal axes along which the Qwen 3B responses were better than the Qwen 1.5B, but we did observe several interesting deltas. We summarize them here, and showcase qualitative examples of these differences in Figure A3, Figure A4, and Figure A5.

1. On prompts with verifiable answers (e.g. math, code), we find pairs where Qwen 3B responds correctly but Qwen 1.5B does not (Figure A3).

2. On knowledge-seeking prompts, we find pairs where Qwen 3B responds with more detail (Figure A4).

3. On prompts that admit brief answers, we find pairs where Qwen 3B generates a chain-of-thought, while Qwen 1.5B responds with just the answer (Figure A5).

Note that these deltas *are not exhaustive*; we simply highlight a few here as interesting examples to motivate future work. In particular, we believe it would be exciting to further characterize what semantic deltas exist in preference data and how the deltas translate to downstream model behavior after training.

## E.3 Quantitative Statistics

We computed the following statistics for each weak dataset used in our core post-training experiments (Table 3) using the Tülu-3-8B-SFT tokenizer (i.e. the base SFT model's tokenizer):

- **Average token length** of chosen and rejected responses

- **Vocabulary diversity**, measured as the number of unique 1-gram and 2-gram tokens in chosen and rejected responses

- **Cosine similarity** between chosen and rejected response embeddings, computed using OpenAI's `text-embedding-3-small` API

As a reference point, we also report these statistics for the original Tülu 3 preference dataset. Results are shown in Table A5. Overall, we observe that the chosen responses in our weak preference data are generally (a) shorter and (b) more diverse in vocabulary than the paired rejected responses. In contrast, chosen and rejected responses in the Tülu 3 data exhibit largely similar statistics. We hypothesize that the longer length of rejected responses in our data may be due to degenerate outputs from the small weak models; they sometimes repeat tokens until reaching the maximum generation length. The cosine similarity between chosen and rejected responses is relatively small for all datasets; qualitatively, chosen and rejected responses often differ significantly on a surface semantic level, which may contribute to this low embedding similarity.

| Preference Dataset | Chosen Responses | | | Rejected Responses | | | Cosine Sim. |
| --- | --- | --- | --- | --- | --- | --- | --- |
| | Avg. Len. | Uniq. 1-gram | Uniq. 2-gram | Avg. Len. | Uniq. 1-gram | Uniq. 2-gram | |
| Llama 3.2 3B over 1B | 779.3 | 112,553 | 6,858,616 | 1,041.3 | 109,090 | 5,884,941 | 0.117 |
| Qwen 2.5 1.5B over 0.5B | 776.5 | 110,380 | 5,107,761 | 1,317.8 | 106,900 | 4,626,119 | 0.112 |
| Qwen 2.5 3B over 1.5B | 709.1 | 114,183 | 6,637,312 | 776.5 | 110,380 | 5,107,761 | 0.115 |
| Tülu 3 Preference Data | 443.9 | 119,269 | 10,980,913 | 441.3 | 119,320 | 10,866,352 | 0.117 |

Table A5: Statistics of the chosen and rejected responses in our weak preference datasets, with statistics of responses from the Tülu 3 preference dataset shown for referenc

## F  Delta Learning in Logistic Regression

In this section, we give the full proofs of Theorem 6.1 and Corollary 6.2 along with all intermediate propositions and lemmas.

### F.1  Additional Preliminaries

We begin by stating additional useful results that we take as preliminaries.

**Proposition F.1** (Population Gradient for Naive Preference Loss). *Take covariates $x \sim \mathcal{N}(0, I_d)$ and assign pseudo-labels $y_c, y_r$ using two teacher models $\theta_c, \theta_r \in \mathbb{R}^d$ via the following rule:*

$$y_c = \mathbb{1}\{\langle \theta_c, x \rangle \geq 0\}, \qquad y_r = \mathbb{1}\{\langle \theta_r, x \rangle \geq 0\}.$$

*Then in expectation over the covariate distribution, we have*

$$\mathbb{E}\left[\nabla_\theta \mathcal{L}_{pref}(x, y_c, y_r; \theta)\right] = -\frac{1}{\sqrt{2\pi}}\left(\frac{\theta_c}{\|\theta_c\|_2} - \frac{\theta_r}{\|\theta_r\|_2}\right),$$

$$\mathrm{Cov}\left(\nabla_\theta \mathcal{L}_{pref}(x, y_c, y_r; \theta)\right) \preceq I_d.$$

*Proof.* The naive preference loss $\mathcal{L}_{\mathrm{pref}}$ is

$$\mathcal{L}_{\mathrm{pref}}(x, y_c, y_r; \theta) = -\left[\log p_\theta(y_c|x) - \log p_\theta(y_r|x)\right], \qquad p_\theta(y = 1|x) = \sigma(\langle \theta, x \rangle).$$

For any single fixed preference pair $(x, y_c, y_r)$, the gradient with respect to $\theta$ is

$$\begin{aligned}
\nabla_\theta L_{\mathrm{naive}}(x, y_c, y_r; \theta) &= -\left[\nabla_\theta \log p_\theta(y_c|x) - \nabla_\theta \log p_\theta(y_r|x)\right] \\
&= -\left[(y_c - \sigma(\langle \theta, x \rangle))x - (y_r - \sigma(\langle \theta, x \rangle))x\right] \\
&= -(y_c - y_r)x.
\end{aligned}$$

Taking expectation of the gradient over the covariate distribution $\mathcal{N}(0, I_d)$,

$$\begin{aligned}
\mathbb{E}[\nabla_\theta \mathcal{L}_{\mathrm{pref}}(x, y_c, y_r; \theta)] &= -\left[\mathbb{E}\left[y_c \cdot x\right] - \mathbb{E}\left[y_r \cdot x\right]\right] \\
&= -\left[\mathbb{E}\left[\mathbb{1}\{\langle \theta_c, x \rangle \geq 0\} \cdot x\right] - \mathbb{E}\left[\mathbb{1}\{\langle \theta_r, x \rangle \geq 0\} \cdot x\right]\right].
\end{aligned}$$

Since the function $h(x) = \mathbb{1}\{\langle \theta, x \rangle \geq 0\}$ is weakly differentiable, by Stein's Lemma we have

$$\begin{aligned}
\mathbb{E}\left[\mathbb{1}\{\langle \theta, x \rangle \geq 0\} \cdot x\right] &= \mathbb{E}\left[\nabla_x \mathbb{1}\{\langle \theta, x \rangle \geq 0\}\right] \\
&= \mathbb{E}\left[\theta \cdot \delta(\langle \theta, x \rangle)\right] = \theta \, \mathbb{E}\left[\delta(\langle \theta, x \rangle)\right] \\
&= \frac{1}{\sqrt{2\pi}} \cdot \frac{\theta}{\|\theta\|_2}.
\end{aligned}$$

where $\delta$ denotes the Dirac delta function. The final equality holds since $Z := \langle \theta, x \rangle \sim \mathcal{N}(0, \|\theta\|_2^2)$, and hence $\mathbb{E}\left[\delta(\langle \theta, x \rangle)\right] = p_Z(0) = 1/(\sqrt{2\pi}\|\theta\|_2)$.

We now bound the covariance $\mathrm{Cov}(\nabla_\theta \mathcal{L}_{\mathrm{pref}}) = \mathrm{Cov}((y_c - y_r)x)$. Let $D$ denote the set where $\theta_c$ and $\theta_r$ disagree on decision

$$D = \{x: \mathrm{sgn}\langle \theta_c, x \rangle \neq \mathrm{sgn}\langle \theta_r, x \rangle\}$$

Then since $(y_c - y_r)x = x$ for all $x \in D$ and 0 otherwise, we have

$$\mathbb{E}\left[(\nabla_\theta \mathcal{L}_{\mathrm{pref}})(\nabla_\theta \mathcal{L}_{\mathrm{pref}})^T\right] = \mathbb{E}[(y_c - y_r)^2 xx^T] = \mathbb{E}[xx^T \cdot \mathbb{1}_D(x)] \preceq \mathbb{E}[xx^T] = I_d$$

$$\implies \mathrm{Cov}(\nabla_\theta \mathcal{L}_{\mathrm{pref}}) = \mathbb{E}[(\nabla_\theta \mathcal{L}_{\mathrm{pref}})(\nabla_\theta \mathcal{L}_{\mathrm{pref}})^T] - \mathbb{E}[\nabla_\theta \mathcal{L}_{\mathrm{pref}}] \mathbb{E}[\nabla_\theta \mathcal{L}_{\mathrm{pref}}]^T \preceq I_d$$

as $\mathbb{E}[\nabla_\theta \mathcal{L}_{\mathrm{pref}}] \mathbb{E}[\nabla_\theta \mathcal{L}_{\mathrm{pref}}]^T$ is positive semi-definite. The equality $\mathbb{E}[xx^T] = I_d$ holds from the assumption that our data is drawn from isotropic Gaussian. $\square$

**Lemma F.2** (Vector Bernstein-Freedman). *Let* $(\mathcal{F})_{k \geq 0}$ *be a filtration and let*

$$\{Y_k : k = 0, 1, 2, \dots\}$$

*be an* $\mathbb{R}^d$*-valued martingale adapted to it. Denote the difference sequence as*

$$X_k := Y_k - Y_{k-1}, \quad k \geq 1,$$

*and assume that the difference sequence is uniformly bounded:*

$$\|X_k\|_2 \leq R \quad \text{almost surely for every } k \geq 1$$

*At any finite horizon n, we define the predictable quadratic variation as*

$$\sigma_n^2 := \sum_{k=1}^n \mathbb{E}_{k-1} \|X_k\|_2^2.$$

*Let* $S_n$ *denote the partial sum of differences up to the horizon,* $S_n = \sum_{k=1}^n X_k$*. Then for every* $t \geq 0$*,*

$$\Pr\left[\|S_n\|_2 \geq t\right] \leq (d+1) \exp -\frac{-t^2/2}{\sigma_n^2 + Rt/3}.$$

*Equivalently, for any failure probability* $\delta \in (0, 1)$*, then with probability at least* $1 - \delta$

$$\|S_n\|_2 \leq \sqrt{2\sigma_n^2 \ln \frac{d+1}{\delta}} + \frac{2R}{3} \ln \frac{d+1}{\delta}.$$

*Proof.* This is a restatement of Theorem 1.6 from Tropp (2012) specialized to $d \times 1$ vectors in $\mathbb{R}^d$. To get the equivalent high probability bound statement, simply set $\Pr\left[\|S_n\|_2 \geq t\right] \leq \delta$ and solve for $t$. $\square$

## F.2 Full Proof of Theorem 6.1

We build up to the final proof via a series of propositions.

**Proposition F.3.** *Take* $a, b \in \mathbb{R}^d$ *and take* $\theta^*$ *to be a unit vector in* $\mathbb{R}^d$*. Define the function*

$$f(t) := \cos(a + bt, \theta^*) = \frac{\langle a + bt, \theta^* \rangle}{\|a + bt\|_2}. \tag{4}$$

*Then the derivative at* $t = 0$ *satisfies the following identity:*

$$f'(0) \cdot \|a\|_2 = \langle \mathrm{Proj}_{a^\perp}(b), \theta^* \rangle. \tag{5}$$

*This can also be expressed as*

$$f'(0) \cdot \|a\|_2 = \langle b, \theta^* \rangle \left(1 - \frac{\langle a, \theta^* \rangle^2}{\|a\|_2^2}\right) - \frac{\langle a, \theta^* \rangle}{\|a\|_2^2} \left(\langle \mathrm{Proj}_{(\theta^*)^\perp}(a), \mathrm{Proj}_{(\theta^*)^\perp}(b) \rangle\right). \tag{6}$$

*Hence,* $f'(0) > 0$ *is equivalent to either of the above expressions being positive.*

*Proof.* This proceeds with direct calculation. Write $\ell(t) = a + bt$. By quotient rule, the derivative of $f$ is

$$f'(t) = \frac{d}{dt}\left[\frac{\langle \ell(t), \theta^* \rangle}{\|\ell(t)\|_2}\right] = \frac{\|\ell(t)\|_2^2 \cdot \langle b, \theta^* \rangle - \langle \ell(t), \theta^* \rangle \cdot \langle \ell(t), b \rangle}{\|\ell(t)\|_2^3}.$$

Evaluating at $t = 0$ gives

$$f'(0) = \frac{\|a\|_2^2 \langle b, \theta^* \rangle - \langle a, \theta^* \rangle \langle a, b \rangle}{\|a\|_2^3}.$$

Now write the projection of $b$ onto the hyperplane perpendicular to $a$ as

$$\mathrm{Proj}_{a^\perp}(b) := b - \frac{\langle a, b \rangle}{\|a\|_2^2} a,$$

so then

$$\langle \mathrm{Proj}_{a^\perp}(b), \theta^* \rangle = \langle b, \theta^* \rangle - \frac{\langle a, b \rangle}{\|a\|_2^2} \langle a, \theta^* \rangle = f'(0) \cdot \|a\|_2,$$

yielding the first identity. We further decompose $a, b$ into their components parallel to and orthogonal to $\theta^*$ as

$$a = \langle a, \theta^* \rangle \theta^* + \mathrm{Proj}_{(\theta^*)^\perp}(a), \qquad b = \langle b, \theta^* \rangle \theta^* + \mathrm{Proj}_{(\theta^*)^\perp}(b).$$

Simplifying $\langle a, b \rangle$ with this decomposition, we have

$$\langle a, b \rangle = \langle a, \theta^* \rangle \langle b, \theta^* \rangle + \langle \mathrm{Proj}_{(\theta^*)^\perp}(a), \mathrm{Proj}_{(\theta^*)^\perp}(b) \rangle.$$

Substituting this into the expression for $\langle \mathrm{Proj}_{a^\perp}(b), \theta^* \rangle$ yields that

$$f'(0) \cdot \|a\|_2 = \langle \mathrm{Proj}_{a^\perp}(b), \theta^* \rangle$$
$$= \langle b, \theta^* \rangle \left(1 - \frac{\langle a, \theta^* \rangle^2}{\|a\|_2^2}\right) - \frac{\langle a, \theta^* \rangle}{\|a\|_2^2} \left(\langle \mathrm{Proj}_{(\theta^*)^\perp}(a), \mathrm{Proj}_{(\theta^*)^\perp}(b) \rangle\right),$$

yielding the second identity. $\qquad\square$

**Proposition F.4.** *Take $f$ as defined in [Proposition F.3], $f(t) := \cos(a + bt, \theta^*)$. Then for all $t$,*

$$|f''(t)| \leq \frac{2}{\sqrt{3}} \cdot \frac{\|b\|_2^2}{\|a + bt\|_2^2}. \tag{7}$$

*Proof.* This bound can be shown by some careful algebra. We rewrite $f$ in terms of two intermediary functions defined as

$$r(t) := \|l(t)\|_2, \qquad u(t) := l(t)/r(t),$$

so that $f = \langle u(t), \theta^* \rangle$. Then since $\theta^*$ is a fixed unit vector we have

$$f''(t) = \langle u''(t), \theta^* \rangle \implies |f''(t)| \leq \|u''(t)\|_2.$$

So it suffices to bound $u''(t)$. With some calculus, we compute

$$u'(t) = \frac{br(t)^2 - l(t)\langle l(t), b \rangle}{r(t)^3}.$$

Differentiating again with quotient rule,

$$u''(t) = \frac{r(t)^3 \cdot \left(2b\langle l(t), b \rangle - b\langle l(t), b \rangle - l(t)\langle b, b \rangle\right) - \left(br(t)^2 - l(t)\langle l(t), b \rangle\right) \cdot 3r(t)\langle l(t), b \rangle}{r(t)^6}$$
$$= \frac{-2b\langle l(t), b \rangle r(t)^3 - l(t)\langle b, b \rangle r^3(t) + 3l(t)\langle l(t), b \rangle^2 r(t)}{r(t)^6}$$
$$= b \cdot \frac{-2\langle l(t), b \rangle}{r(t)^3} + l(t) \cdot \left(-\frac{\langle b, b \rangle}{r(t)^3} + \frac{3\langle l(t), b \rangle^2}{r(t)^5}\right).$$

To simplify further, we decompose $b$ into constituent parts parallel to and perpendicular to $l(t)$:

$$b_{\parallel} := \text{Proj}_{l(t)}(b) = \left( \frac{\langle l(t), b \rangle}{r(t)^2} \right) l(t), \qquad b_{\perp} := b - b_{\parallel}.$$

Then since $b = b_{\parallel} + b_{\perp}$ we can reduce $u''(t)$ as

$$u''(t) = \left( l(t) \cdot \frac{-2\langle l(t), b \rangle^2}{r(t)^5} + b_{\perp} \cdot \frac{-2\langle l(t), b \rangle}{r(t)^3} \right) + l(t) \cdot \left( -\frac{\langle b, b \rangle}{r(t)^3} + \frac{3\langle l(t), b \rangle^2}{r(t)^5} \right)$$

$$= b_{\perp} \cdot \frac{-2\langle l(t), b \rangle}{r(t)^3} - l(t) \cdot \frac{\langle b, b \rangle - \langle l(t), b \rangle^2/r(t)^2}{r(t)^3}.$$

So now we've expressed $u''(t)$ in terms of two orthogonal components, $b_{\perp} \perp l(t)$. As such, the squared norms of the components add:

$$\|u''(t)\|_2^2 = \frac{4\langle l(t), b \rangle^2 \|b_{\perp}\|_2^2}{r(t)^6} + \frac{\left( \|b\|_2^2 - \langle l(t), b \rangle^2/r(t)^2 \right)^2}{r(t)^4} \qquad \left( \|l(t)\|_2^2 = r(t)^2 \right)$$

Now let $\theta(t)$ denote the angle between $b$ and $l(t)$, and observe that $c(t) := \cos(\theta(t)) = \frac{\langle l(t), b \rangle}{r(t)\|b\|_2}$. Then we have

$$\|u''(t)\|_2^2 = \frac{4c(t)^2 \|b\|_2^2 \cdot (1 - c(t)^2) \|b\|_2^2}{r(t)^4} + \frac{\left( \|b\|_2^2 - c(t)^2 \|b\|_2^2 \right)^2}{r(t)^4}$$

$$= \frac{\|b\|_2^4}{r(t)^4} \cdot \left[ 4c(t)^2(1 - c(t)^2) + (1 - c(t)^2)^2 \right]$$

$$= \frac{\|b\|_2^4}{r(t)^4} \cdot \left[ -3c(t)^4 + 2c(t)^2 + 1 \right].$$

By construction, for any $t$ we have $c(t) \in [-1, 1]$. One can check that

$$|f''(t)|^2 \leq \|u''(t)\|_2^2 \leq \frac{\|b\|_2^4}{r(t)^4} \cdot \sup_{x \in [-1,1]} \left[ -3x^4 + 2x^2 + 1 \right] \leq \frac{\|b\|_2^4}{r(t)^4} \cdot (4/3).$$

Taking square roots of both sides gives the desired bound. □

**Proposition F.5.** *Assume the delta learning setup of Section 6.1 and Theorem 6.1, and suppose we train with the population update rule*

$$\overline{\theta_{t+1}} \leftarrow \overline{\theta_t} - \eta \, \mathbb{E}_{x \sim \mathcal{N}(0, I_d)} \left[ \nabla_\theta \mathcal{L}_{pref}(x, y_c, y_r; \overline{\theta_t}) \right]. \tag{8}$$

*Then if Condition C1 holds so that $\kappa > 0$ (as defined in Theorem 6.1), training with learning rate $\eta$ for a total update horizon of*

$$H^* := \eta T = \frac{\kappa \|\theta_0\|_2}{4 \|v_\Delta\|_2^2} \tag{9}$$

*yields a final iterate $\overline{\theta_T}$ whose cosine alignment with the ideal parameters $\theta^*$ improves over the alignment of the initial student $\theta_0$ by a margin of at least $\Gamma$:*

$$\Gamma := \kappa/50 = \Theta(\kappa), \tag{10}$$

$$\cos(\overline{\theta_T}, \theta^*) > \cos(\theta_0, \theta^*) + \Gamma. \tag{11}$$

*Proof.* As discussed in our sketch in Section 6.2, by Proposition F.1 training with population gradients amounts to tracing various points along a parametric ray $\ell : \mathbb{R}_{\geq 0} \mapsto \mathbb{R}^d$,

$$\ell(\lambda) := \theta_0 + \lambda v_\Delta, \qquad v_\Delta := \theta_c / \|\theta_c\|_2 - \theta_r / \|\theta_r\|_2. \tag{12}$$

Define $f$ to be a map measuring the student's performance over the course of training:

$$f(\lambda) := \cos(\ell(\lambda), \theta^*) = \frac{\langle \ell(\lambda), \theta^* \rangle}{\|\ell(\lambda)\|_2}. \tag{13}$$

Specializing Proposition F.3 to our setup, we have that

$$f'(0) \cdot \|\theta_0\|_2 = \langle \text{Proj}_{\theta_0^\perp}(v_\Delta), \theta^* \rangle \tag{14}$$

$$= (\alpha_c - \alpha_r)(1 - \alpha_0^2) - \alpha_0 \langle \text{Proj}_{(\theta^*)^\perp}(\theta_0/\|\theta_0\|_2), \text{Proj}_{(\theta^*)^\perp}(v_\Delta) \rangle =: \kappa. \tag{15}$$

So $f'(0) \iff \kappa > 0$, and $f'(0) > 0$ is a sufficient condition for training with population updates to yield an improvement. This is exactly Condition C1 from Theorem 6.1. Assuming this holds, we can quantify the magnitude of the gain after training for some $T$ steps. By a second-order Taylor expansion of $f$, the total improvement $\Gamma$ can be bounded as

$$\Gamma := \cos(\overline{\theta_T}, \theta^*) - \cos(\theta_0, \theta^*) = f(\eta T) - f(0) \geq \eta T f'(0) - \frac{L}{2}(\eta T)^2, \tag{16}$$

where $L$ is a bound on the second derivative $|f''(\lambda)|$ derived in Proposition F.4:

$$L := \sup_{\lambda \in [0, \eta T]} |f''(\lambda)| = \frac{2}{\sqrt{3}} \frac{\|v_\Delta\|_2^2}{\|\theta_0 + \lambda v_\Delta\|_2^2}. \tag{17}$$

To simplify $L$ further, observe that $|\kappa| \leq 2\|v_\Delta\|_2$:

$$|\kappa| \leq \left| \langle v_\Delta, \theta^* \rangle (1 - \alpha_0^2) \right| + \left| \alpha_0 \langle \text{Proj}_{(\theta^*)^\perp}(\theta_0/\|\theta_0\|_2), \text{Proj}_{(\theta^*)^\perp}(v_\Delta) \rangle \right| \tag{18}$$

$$\leq \|v_\Delta\|_2 + \left\| \text{Proj}_{(\theta^*)^\perp}(v_\Delta) \right\|_2 \leq 2\|v_\Delta\|_2. \tag{19}$$

So then we have for all $\lambda \in [0, \eta T]$ that

$$\|\theta_0 + \lambda v_\Delta\|_2 \geq \|\theta_0\|_2 - \lambda \|v_\Delta\|_2 \geq \|\theta_0\|_2 - \left( \frac{2\|v_\Delta\|_2 \|\theta_0\|_2}{4\|v\|_2^2} \right) \|v_\Delta\|_2 = \frac{1}{2}\|\theta_0\|_2, \tag{20}$$

$$\implies L = \frac{8}{\sqrt{3}} \frac{\|v_\Delta\|_2^2}{\|\theta_0\|_2^2}. \tag{21}$$

Now, note that $f'(0) = \kappa/\|\theta_0\|_2$ (Equation 15), and set the total training horizon as in the proposition statement:

$$H^* := \eta T = f'(0) \cdot \frac{\|\theta_0\|_2^2}{4\|v_\Delta\|_2^2} = \frac{\kappa \|\theta_0\|_2}{4\|v_\Delta\|_2^2}. \tag{22}$$

Combining this training horizon with Equation 16 and Equation 21, we have $\eta T = 2f'(0)/(\sqrt{3}L)$, so

$$\Gamma = \frac{2f'(0)^2}{\sqrt{3}L} - \frac{L}{2} \cdot \frac{4f'(0)^2}{3L^2} = \left( \frac{2}{\sqrt{3}} - \frac{2}{3} \right) \frac{f'(0)^2}{L} = \frac{\sqrt{3}}{8} \left( \frac{2}{\sqrt{3}} - \frac{2}{3} \right) \frac{\kappa^2}{\|v_\Delta\|_2^2}. \tag{23}$$

Simplifying with $\|v_\Delta\|_2 \leq \|\theta_c\|_2 + \|\theta_r\|_2 = 2$ yields the desired claim. $\qquad \square$

**Proposition F.6.** *Assume the delta learning setup of Section 6.1 and Theorem 6.1. Then the student iterates $\theta_t$ trained with empirical mini-batch SGD do not deviate too much from the exact iterates $\overline{\theta}_t$ trained with population gradients (Equation 8). Formally, for any failure probability $\delta \in (0, 1)$, if the ambient dimension satisfies $d > \frac{1}{4}\ln(2BT/\delta)$, then with probability at least $1 - \delta$*

$$\|\theta_T - \overline{\theta}_T\|_2 \leq \eta \left[ \sqrt{\frac{2dT}{B} \ln \frac{d+1}{\delta/2}} + 4\sqrt{d} \ln \frac{d+1}{\delta/2} \right] = \eta \, \widetilde{O}\left( \sqrt{dT/B} + \sqrt{d} \right). \tag{24}$$

*Proof.* This follows via a martingale concentration argument. Write the empirical mini-batch gradient at each timestep $t$ as

$$g_t = \frac{1}{B} \sum_{i=1}^{B} \left[ \nabla_\theta \mathcal{L}_{\text{pref}}\left(x^{(t,i)}, y_c^{(t,i)}, y_r^{(t,i)}\right) \right] \tag{25}$$

and define the *error vector*

$$\zeta_t := g_t - \mathbb{E}[g_t], \tag{26}$$

where $\mathbb{E}[g_t] = -v_\Delta$ up to a constant scaling factor absorbed into $\eta$. Each $\zeta_t$ is a random variable (over the draws of the mini-batch) that measures how much the empirical gradient $g_t$ differs from the population gradient. Introducing $\zeta_t$ allows us to rewrite the empirical SGD updates in terms of the exact iterates $\overline{\theta_t}$:

$$\theta_{t+1} = \theta_t - \eta g_t = (\theta_t - \eta \, \mathbb{E}[g_t]) - \eta \zeta_t = \overline{\theta_{t+1}} - \eta \zeta_t. \tag{27}$$

Unrolling these updates across $T$ steps,

$$\theta_T = \overline{\theta_T} - \eta \sum_{t=1}^{T} \zeta_t \qquad \left(= \text{deterministic "backbone" + stochastic deviation}\right). \tag{28}$$

We now bound the stochastic deviation $\eta \sum_{i=1}^{T} \zeta_t$. By construction, the sequence $\{\zeta_1, \dots, \zeta_T\}$ is a martingale difference sequence with $\mathbb{E}[\zeta_t] = 0$. Hence, their cumulative sum can be bounded via a vector Bernstein-Freedman inequality (Lemma F.2). We verify the necessary assumptions for Lemma F.2 by bounding $\|\zeta_t\|_2$ and $\sum_{t=1}^{T} \mathbb{E} \|\zeta_t\|_2^2$. To bound $\|\zeta_t\|_2$, observe that $x^{(t,i)} \sim \mathcal{N}(0, I_d)$, so then $\left\|x^{(t,i)}\right\|_2^2$ follows a chi-squared distribution with $d$ degrees of freedom. Standard tail bounds due to Laurent & Massart (2000) give that

$$\Pr\left(\left\|x^{(t,i)}\right\|_2 \geq 4\sqrt{d}\right) \leq e^{-4d}. \tag{29}$$

By a union bound, the event that *any* covariate observed throughout training exceeds this bound occurs with probability at most $\delta_1 = TB \exp(-4d)$. So if $d$ exceeds the stated threshold, up to a $\delta_1 = \delta/2$ failure probability we can condition the rest of our argument on the "good" event

$$\mathcal{E} := \left\{ \mathbb{1}\left\{ \left\|x^{(t,i)}\right\|_2 \leq 4\sqrt{d} \right\} \forall i \, \forall t \right\}. \tag{30}$$

Applying triangle inequality then gives

$$\|\zeta_t\|_2 \leq \|g_t\|_2 + \|v_\Delta\|_2 \leq \frac{1}{B} \sum_{i=1}^{B} \left\|x^{(t,i)}\right\|_2 + \|v_\Delta\|_2 \leq 4\sqrt{d} + \|v_\Delta\|_2 \leq 6\sqrt{d}. \tag{31}$$

The final inequality holds because $\|v_\Delta\|_2 \leq \|\theta_c\|_2 + \|\theta_r\|_2 = 2$. We next bound the cumulative second moments $\sum_{t=1}^{T} \mathbb{E} \|\zeta_t\|_2^2$. Observe that

$$\text{Cov}(\zeta_t) = \text{Cov}(g_t) = \frac{1}{B^2} \sum_{i=1}^{B} \text{Cov}\left(\nabla_\theta \mathcal{L}_{\text{pref}}\left(x_i^{(t)}, y_{c,i}^{(t)}, y_{r,i}^{(t)}\right)\right) \preceq \frac{1}{B} I_d, \qquad \text{(Proposition F.1)} \tag{32}$$

so then the second moment is bounded as

$$\mathbb{E}\left[\|\zeta_t\|_2^2\right] \leq \frac{d}{B} \implies \sum_{i=1}^{T} \mathbb{E}\left[\|\zeta_t\|_2^2\right] \leq \frac{dT}{B}. \tag{33}$$

Then by Lemma F.2, for any $\delta_2 \in (0, 1)$ we have with probability at least $1 - \delta_2$

$$\left\|\sum_{i=1}^{T} \zeta_t\right\|_2 \leq \sqrt{\frac{2dT}{B} \ln \frac{d+1}{\delta_2}} + 4\sqrt{d} \ln \frac{d+1}{\delta_2}. \tag{34}$$

Combining this with Equation 28 and setting $\delta_2 = \delta/2$ yields that the desired claim. $\qquad \square$

We are now ready to put everything together.

**_Proof of Theorem 6.1._** We want to show that the early-stopped $T$-th student iterate $\theta_T$ achieves higher performance than the initial student $\theta_0$,

$$\cos(\theta_T, \theta^*) - \cos(\theta_0, \theta^*) \geq \Theta(\kappa^2) > 0.$$

Using a triangle inequality, we break this down as

$$\cos(\theta_T, \theta^*) - \cos(\theta_0, \theta^*) \geq \underbrace{\left[\cos(\overline{\theta_T}, \theta^*) - \cos(\theta_0, \theta^*)\right]}_{\text{deterministic ideal gain}} - \underbrace{\left|\cos(\overline{\theta_T}, \theta^*) - \cos(\theta_T, \theta^*)\right|}_{\text{stochastic deviation from ideal}}. \quad (35)$$

By Proposition F.5, if Condition C1 in the theorem statement holds and we fix the total training horizon $H^* = \eta T = (\kappa \|\theta_0\|_2) / (4 \|v_\Delta\|_2^2)$, then we are guaranteed a deterministic gain in cosine similarity of at least $\Gamma \geq \Theta(\kappa^2)$. To bound the stochastic error, we control $\|\overline{\theta_T} - \theta_T\|_2$ (Proposition F.6) and rely on the continuity of cosine similarity. A straightforward gradient computation shows that for all fixed $R > 0$, the map $f_u(\theta) = \langle \theta, u \rangle / \|\theta\|_2$ is $(1/R)$-Lipschitz on the domain $\{\theta \in \mathbb{R}^d, \|\theta\|_2 \geq R\}$. In our setting, training for horizon $H^*$ already guarantees that $\|\overline{\theta_T}\|_2 \geq \frac{1}{2}\|\theta_0\|_2$ (Equation 20), so

$$\|\theta_T\|_2 \geq \|\overline{\theta_T}\|_2 - \|\overline{\theta_T} - \theta_T\|_2 \geq \frac{1}{2}\|\theta_0\|_2 - \|\overline{\theta_T} - \theta_T\|_2. \quad (36)$$

Thus, if we control the distance such that

$$\|\overline{\theta_T} - \theta_T\|_2 \leq \frac{1}{8}\Gamma\|\theta_0\|_2, \quad (37)$$

then we have $\|\theta_T\|_2 \geq \frac{1}{4}\|\theta_0\|_2$ (as $0 < \Gamma \leq 1$) and consequently

$$\left|\cos(\overline{\theta_T}, \theta^*) - \cos(\theta_T, \theta^*)\right| \leq \frac{4}{\|\theta_0\|_2}\|\overline{\theta_T} - \theta_T\|_2 = \frac{\Gamma}{2}. \quad (38)$$

So then training $\theta_0$ with delta learning yields an improvement in cosine similarity of at least $\Gamma/2 = \Theta(\kappa^2)$ as claimed. We end by analyzing what the training hyperparameters $\eta, T, B$ need to be set as to control this distance. Rewriting $T = H^*/\eta$ and combining Proposition F.6 with Equation 37,

$$\|\overline{\theta_T} - \theta_T\|_2 \lesssim \eta \left[ \frac{1}{\sqrt{\eta}} \underbrace{\sqrt{\frac{2dH^*}{B} \ln(d/\delta)}}_{=: a} + \underbrace{4\sqrt{d}\ln(d/\delta)}_{=: b} \right] = a\eta^{1/2} + b\eta \leq \frac{1}{8}\Gamma\|\theta_0\|_2. \quad (39)$$

We can rewrite Equation 39 as a quadratic inequality in $s = \sqrt{\eta}$:

$$bs^2 + as - \frac{1}{8}\Gamma\|\theta_0\|_2 \leq 0. \quad (40)$$

Because both $a, b > 0$, $\eta = s^2$ satisfies Equation 39 so long as we simultaneously have

$$bs^2 \leq \frac{1}{16}\Gamma\|\theta_0\|_2 \quad \text{and} \quad as \leq \frac{1}{16}\Gamma\|\theta_0\|_2.$$

Both are immediately satisfied if we set $\eta$ as

$$\eta := \min\left\{ \frac{\Gamma\|\theta_0\|_2}{16b}, \frac{\Gamma^2\|\theta_0\|_2^2}{256a^2} \right\}. \quad (41)$$

Plugging in the definitions of $a, b, \Gamma$ into Equation 41 and choosing $B = \Theta(d)$, we have (modulo some fixed scaling constants)

$$\eta = \min\left\{ \frac{\kappa^2\|\theta_0\|_2}{\sqrt{d}\ln(d/\delta)}, \frac{\kappa^3 B\|\theta_0\|_2}{d\|v_\Delta\|_2^2 \ln(d/\delta)} \right\} = \widetilde{\Theta}\left(\kappa^2\|\theta_0\|_2 \cdot \min\left\{1/\sqrt{d}, \kappa/\|v_\Delta\|_2^2\right\}\right). \quad (42)$$

So then the number of training steps $T$ can be expressed as

$$T = H^*/\eta = \widetilde{\Theta}\left(\frac{1}{\kappa \|v_\Delta\|_2^2} \cdot \max\left\{\sqrt{d}, \|v_\Delta\|_2^2/\kappa\right\}\right). \tag{43}$$

Finally, our earlier use of Proposition F.6 requires that

$$d > \frac{1}{4}\ln(2BT/\delta). \tag{44}$$

But we set $B = \Theta(d)$ and $T = O(\sqrt{d})$, so the right-hand side grows only logarithmically with $d$; this condition is trivially satisfied for any reasonably large $d$. One can check this with some algebra:

$$\frac{1}{4}\ln(2BT/\delta) \lesssim \frac{1}{4}\ln\left(\frac{2d}{\delta} \cdot \frac{\max\left\{\sqrt{d}, \|v_\Delta\|_2^2/\kappa\right\}}{\kappa\|v_\Delta\|_2^2} \cdot \ln(d/\delta)\right) \tag{45}$$

$$\leq \frac{1}{4}\ln\left(\frac{2d^2}{\delta^2} \cdot \frac{\sqrt{d}(1 + \|v_\Delta\|_2^2/\kappa)}{\kappa\|v_\Delta\|_2^2}\right) \leq \frac{1}{4}\left[3\ln(d) + \ln\left(\frac{\kappa + \|v_\Delta\|_2^2}{\delta^2\kappa\|v_\Delta\|_2^2}\right)\right] \tag{46}$$

$$\leq \frac{3}{4}d + \frac{1}{4}\ln\left(\frac{\kappa + \|v_\Delta\|_2^2}{\delta^2\kappa\|v_\Delta\|_2^2}\right) \leq d. \tag{47}$$

So as long as

$$d \gtrsim \ln\left(\frac{\kappa + \|v_\Delta\|_2^2}{\delta^2\kappa\|v_\Delta\|_2^2}\right) \tag{48}$$

then the total failure probability is at most $\delta$, and running delta learning with SGD improves $\theta_0$ by $\Theta(\kappa^2)$ as claimed. □

### F.3 Full Proof of Corollary 6.2

The desired corrolary can be formally stated as follows:

**Corollary F.7** (Formalization of Corollary 6.2). *In high dimensions, most pairs of teacher models with a performance gap suffice to improve the student via delta learning. Suppose we randomly sample two teacher models $\theta_c, \theta_r$ uniformly over the unit sphere, conditional on their cosine similarity with the optimal model:*

$$\theta_c \sim \text{Uniform}\{\theta \in S^{d-1} \mid \cos(\theta, \theta^*) = \alpha_c\}, \quad \theta_r \sim \text{Uniform}\{\theta \in S^{d-1} \mid \cos(\theta, \theta^*) = \alpha_r\},$$

*and train student $\theta_0$ following the setup from Theorem 6.1. For any $\delta \in (0,1)$, define the threshold*

$$d^* = 2\ln\frac{4}{\delta} \cdot \left(\frac{|\alpha_0| \|\theta_0\|_2 \left(\sqrt{1 - \alpha_c^2} + \sqrt{1 - \alpha_r^2}\right)}{(\alpha_c - \alpha_r)(1 - \alpha_0^2)}\right)^2 + 1. \tag{49}$$

*Then whenever $d > d^*$, with probability at least $1 - \delta$ Condition C1 holds and by Theorem 6.1 training strictly improves the student with high probability.*

*Remark F.1.* The expected improvement, on the order of $\Theta(\kappa^2)$, grows with the magnitude of the teachers' performance delta $(\alpha_c - \alpha_r)$ but shrinks as the student's initial performance $\alpha_0$ increases. This theoretical result aligns with our empirical results in Section 5: the quality delta between chosen and rejected responses is a strong predictor of downstream preference tuning performance (albeit only up to a point beyond which gains saturate; language model tuning is more complex than the logistic-regression setup assumed here).

*Remark F.2.* The dimension threshold $d^*$ is mild. Say the initial student has unit parameters $\theta_0$ and is 80% accurate. If the teachers $\theta_c$, $\theta_r$ are sampled to be 70% and 60% accurate respectively, then with $d > 2000$, at least 90% of all such teacher pairs suffice to improve $\theta_0$.

To prove Corollary F.7, we rely on a standard concentration bound of inner products on a sphere:

**Lemma F.8** (Sphere-concentration Bound). *Fix an arbitrary vector $v_0 \in e_1^\perp \subset R^d$, and draw $u_c, u_r$ uniformly and independently from the unit sphere $S^{d-2} \subset e_1^\perp$. For some constants $\alpha_c, \alpha_r \in \mathbb{R}$, set*

$$v_c = \sqrt{1 - \alpha_c^2}\, u_c, \qquad v_r = \sqrt{1 - \alpha_r^2}\, u_r, \qquad v_\Delta = v_c - v_r.$$

*Then for any $\delta \in (0, 1)$, with probability at least $1 - \delta$,*

$$|\langle v_0, v_\Delta \rangle| \;\leq\; \|v_0\|_2 \left( \sqrt{1 - \alpha_c^2} + \sqrt{1 - \alpha_r^2} \right) \sqrt{\frac{2}{d-1} \ln \frac{4}{\delta}}$$

*Proof.* We decompose the inner product and apply Lévy's concentration on each term. We have

$$\langle v_0, v_\Delta \rangle = \sqrt{1 - \alpha_c^2} \langle v_0, u_c \rangle - \sqrt{1 - \alpha_r^2} \langle v_0, u_r \rangle.$$

For either $u \in \{u_c, u_r\}$ the map $f(u) = \langle v_0, u \rangle$ is $\|v_0\|_2$-Lipschitz in the Euclidean metric. Since $\mathbb{E}\langle v_0, u \rangle = 0$, Lévy's concentration on the sphere gives, for any $\epsilon > 0$,

$$\Pr\big(|\langle v_0, u \rangle| \geq \epsilon\big) \;\leq\; 2\exp\Big(-\frac{(d-1)\,\epsilon^2}{2\|v_0\|^2}\Big).$$

Now set $\epsilon$ so that the right hand side equals $\delta/2$; explicitly,

$$\epsilon \;=\; \|v_0\| \sqrt{\frac{2}{d-1} \ln \frac{4}{\delta}}.$$

By a union bound, with probability at least $1 - \delta$ we have $\{|\langle v_0, u_c \rangle| \leq \epsilon\}$ and $\{|\langle v_0, u_r \rangle| \leq \epsilon\}$. So by a triangle inequality we have

$$\big|\langle v_0, v_\Delta \rangle\big| \leq \sqrt{1 - \alpha_c^2}|\langle v_0, u_c \rangle| + \sqrt{1 - \alpha_r^2}|\langle v_0, u_r \rangle| \leq \left( \sqrt{1 - \alpha_c^2} + \sqrt{1 - \alpha_r^2} \right) \epsilon.$$

Substituting the specified $\epsilon$ gives the displayed bound. $\qquad\square$

**Proof of Corollary F.7.** The claim is that Condition C1 holds with high probability in sufficiently high dimensions. Since both $\theta_c, \theta_r$ are randomly drawn and uncorrelated in all components orthogonal to $\theta^*$, standard sphere concentration bounds show that the "bad" noise

$$\langle \text{Proj}_{(\theta^*\perp)}(\tilde{\theta}_0), \text{Proj}_{(\theta^*\perp)}(v_\Delta) \rangle \tag{50}$$

vanishes in high dimensions. In particular, given any fixed failure probability $\delta \in (0, 1)$, a direct application of Lemma F.8 yields that with probability at least $1 - \delta$

$$\left| \langle \text{Proj}_{(\theta^*\perp)}(\tilde{\theta}_0), \text{Proj}_{(\theta^*\perp)}(v_\Delta) \rangle \right| \leq \|\theta_0\|_2 \left( \sqrt{1 - \alpha_c^2} + \sqrt{1 - \alpha_r^2} \right) \sqrt{\frac{2}{d-1} \ln \frac{4}{\delta}}. \tag{51}$$

Hence Condition C1 holds with the same probability whenever

$$\frac{(\alpha_c - \alpha_r)\,(1 - \alpha_0^2)}{|\alpha_0|} > \|\theta_0\|_2 \left( \sqrt{1 - \alpha_c^2} + \sqrt{1 + \alpha_r^2} \right) \sqrt{\frac{2}{d-1} \ln \frac{4}{\delta}}. \tag{52}$$

Or equivalently, whenever

$$|\alpha_0|\, \|\theta_0\|_2 \left( \sqrt{1 - \alpha_c^2} + \sqrt{1 + \alpha_r^2} \right) < (\alpha_c - \alpha_r)(1 - \alpha_0^2) \sqrt{\frac{d-1}{2\ln(4/\delta)}}. \tag{53}$$

In particular, for any fixed $\delta$, the right-hand side grows like $\sqrt{d}$ as $d \to \infty$, so Condition C1 is easily satisfied. Specifically, we simply need

$$d \;>\; d^* := 2\ln(4/\delta) \cdot \left( \frac{|\alpha_0|\, \|\theta_0\|_2 \left( \sqrt{1 - \alpha_c^2} + \sqrt{1 - \alpha_r^2} \right)}{(\alpha_c - \alpha_r)(1 - \alpha_0^2)} \right)^2 + 1. \tag{54}$$

$\qquad\square$

## G Experiment Details

### G.1 Shared training and hyperparameter details

For all training, we generally sweep hyperparameters near the defaults suggested by recent work (Lambert et al., 2024; Hu et al., 2024). We use a cosine annealing learning rate schedule with a warmup ratio of 0.03. We use an AdamW optimizer with ($\beta_1 = 0.9, \beta_2 = 0.95$) and no weight decay following (Ivison et al., 2024). For all DPO tuning, we use length-normalization in the loss, which Lambert et al. (2024) suggests to generally work better. We use batch size 32 for DPO tuning and batch size 256 for SFT. We train all DPO models for exactly one epoch, sweeping learning rate and DPO $\beta$. For all SFT experiments, we sweep epochs and learning rate; see each experiment subsection below for exact ranges. We use gradient checkpointing, DeepSpeed (Rasley et al., 2020), and FlashAttention2 (Dao, 2024) to improve training efficiency. We use code from the OpenRLHF Github repository (Hu et al., 2024) to train all of our models.

### G.2 Pilot Study on ULTRAFEEDBACK-WEAK

**Data and filtering.** The original ULTRAFEEDBACK dataset (Cui et al., 2023) is a popular preference dataset constructed by prompting a set of LLMs with diverse prompts and then scoring the responses using a much stronger judge model (GPT-4). For each prompt $x$, we form preference pairs $(x, y_c, y_r)$ by selecting the highest-scoring response $y_c$ and one lower-scoring response $y_r$.

As of March 28 2025, Llama-3.2-3B-Instruct achieves a LMSYS Chatbot Arena ELO score of 1103 and Llama-3.1-8B-Instruct achieves 1176 ELO. We filter out all responses from the original ULTRAFEEDBACK dataset that were generated by models with higher ELO than 1100. This excludes GPT-4-0613 (1163 ELO), GPT-3.5-Turbo (1106 ELO), and WizardLM-70B (1106 ELO). The best remaining model is Vicuna-33B (1091 ELO); see Table A6 for a full list of remaining models.

| Model | Reference |
|---|---|
| Alpaca-7B | Taori et al. (2023) |
| Bard | https://bard.google.com/ |
| Falcon-40B-Instruct | Almazrouei et al. (2023) |
| Llama-2-13B-Chat | Touvron et al. (2023) |
| Llama-2-70B-Chat | Touvron et al. (2023) |
| Llama-2-7B-Chat | Touvron et al. (2023) |
| MPT-30B-Chat | Team (2023) |
| Pythia-12B | Biderman et al. (2023) |
| StarChat | Tunstall et al. (2023) |
| UltraLM-13B | Ding et al. (2023) |
| UltraLM-65B | Ding et al. (2023) |
| Vicuna-33B | Chiang et al. (2023) |
| WizardLM-7B | Xu et al. (2023) |
| WizardLM-13B | Xu et al. (2023) |

Table A6: Models used to generate the responses in our ULTRAFEEDBACK-WEAK dataset, constructed by filtering out all responses generated by models with a LMSYS Chatbot Arena ELO score above 1100 (i.e., near Llama-3.2-3B-Instruct's ELO) from the original ULTRAFEEDBACK dataset.

**Evaluation.** We evaluate on the eight core benchmarks detailed in Appendix B: MMLU, MATH, GSM8k, IFEval, AlpacaEval 2, TruthfulQA, BigBenchHard, and Codex HumanEval+.

**Training and hyperparameters.** We follow the setup described in Appendix G.1, and further sweep learning rate in {1e-7, 3e-7, 5e-7, 7e-7} and $\beta \in \{5, 10\}$ for DPO training. For SFT, we sweep learning rate in {1e-5, 5e-5, 1e-6} and epochs in {1, 2}.

### G.3 Controlled Experiment: Stylistic Delta in Number of Bold Sections

**Data.** Each prompt $x$ is formed by appending "Include bolded sections in your response." to a prompt from the Tülu 3 SFT dataset (Lambert et al., 2024). To generate each response $y_{k_i}$, we modify the appended instruction into a hard constraint: "Include exactly $k_i$ bolded sections in your response." We generate responses with Llama-3.2-3B-Instruct, and use regular expressions to guarantee correctness. We collect exactly 16384 training data points.

**Hyperparameters.** We follow the setup described in Appendix G.1, and further sweep learning rate in {1e-7, 3e-7, 5e-7, 7e-7} and $\beta \in \{5, 10\}$ for DPO training. For SFT, we sweep learning rate in {1e-5, 5e-5, 1e-6} and epochs in {1, 2}. We select the best hyperparameters based on a held-out validation set.

### G.4 Controlled Experiment: Semantic Delta from a Weaker Model

**Hyperparameters.** We follow the setup described in Appendix G.1, and further sweep learning rate in {3e-7, 1e-7, 5e-8, 1e-8} and $\beta \in \{5, 10\}$ for DPO training. For SFT, we sweep learning rate in {5e-6, 1e-6, 5e-7, 1e-7} and epochs in {1, 2}. We select the best hyperparameters based only on GSM8k accuracy, keeping the remaining evaluations in this controlled experiment held-out.

The learning rates swept here are slightly lower than those used in prior work (Lambert et al., 2024); we found in our preliminary experiments that lower learning rates generally performed better across the board for both DPO and SFT in this setting.

**Evaluation.** We evaluate on the eight core benchmarks detailed in Appendix B: MMLU, MATH, GSM8k, IFEval, AlpacaEval 2, TruthfulQA, BigBenchHard, and Codex HumanEval+.

### G.5 Post-training with Weak Preference Data

**Data.** See Appendix E for details on dataset generation, dataset statistics, and qualitative examples.

**Hyperparameters.** We follow the setup described in Appendix G.1. Hyperparameter tuning is crucial for performant post-training; we carefully tune hyperparameters for each dataset independently, being careful to sweep the same number of hyperparameters for each setting. Following best practice (Lambert et al., 2024; Ivison et al., 2024), we sweep DPO learning rate in {5e-7, 1e-7, 7e-8, 5e-8} and $\beta \in \{5, 10\}$ and select the best checkpoint for each dataset. We further swept dataset size in {100000, 150000, 200000, 264806}, and find that training on a subset of the full dataset (264806 samples) was typically slightly better. Finally, Tülu 3 (Lambert et al., 2024) finds that performance can depend on random seed initialization and hence picks the best run out of multiple seeds; we follow this practice and sweep 5 random seeds on top of our single best hyperparameter and dataset configuration. This yields the numbers for our best setup in Table 3.

**Evaluation.** We evaluate on all eleven benchmarks detailed in Appendix B: MMLU, PopQA, MATH, GSM8k, IFEval, AlpacaEval 2, TruthfulQA, BigBenchHard, DROP, Codex HumanEval, and Codex HumanEval+. We compare directly against the officially released Tülu-3-8B-DPO model and re-run evaluations with the exact same version of the codebase we use to evaluate our own models. We find that, despite using the same evaluation configuration and overall evaluation codebase as Tülu 3 (Appendix B), re-running with the current version slightly improves Tülu-3-8B-DPO's performance numbers compared to the numbers reported in the original paper (Lambert et al., 2024).

### G.6 Analysis Experiments

**Data.** To construct the 21 preference datasets used in our analysis experiments on **delta magnitude** and **chosen response quality**, we use a 100k prompt random subset of the full Tülu 3 dataset.

**Hyperparameters.** We follow the setup described in Appendix G.1. For our analysis experiments on **delta magnitude** and **chosen response quality**, we tune learning rate in $\{$1e-7, 7e-8, 5e-8$\}$ and $\beta \in \{5, 10\}$ when training with DPO. When training with SFT for our **chosen response quality** experiments, we sweep learning rate in $\{$5e-6, 1e-5, 1e-6$\}$ and epochs in $\{1, 2\}$. For our experiments with **GPT-4o annotations** and **OLMo-2-7B-SFT**, we tune hyperparameters as described in Appendix G.5.

**Evaluation.** We evaluate on all eleven benchmarks detailed in Appendix B: MMLU, PopQA, MATH, GSM8k, IFEval, AlpacaEval 2, TruthfulQA, BigBenchHard, DROP, Codex HumanEval, and Codex HumanEval+.

## H Compute Details

All models are trained on either single H100 or A100 nodes. Training an 8B language model with DPO on 100k preference pairs takes approximately 4-6 hours on one H100 node. Supervised finetuning an 8B model on 100k data points takes approximately 2-4 hours on one H100 node.

---

**Llama-3.2-3B-Instruct (no training baseline)**

**User**: ...assuming the graph is Eulerian, determine the number of possible closed trails of length 7 (i.e., the number of Eulerian circuits of length 7) that the guide can choose from, if possible. (Note: A graph is Eulerian if it has an Eulerian circuit, and it must satisfy certain conditions). If the tour guide decides to add an additional attraction to the itinerary, represented by adding a new node and connecting it with edges to 3 existing nodes, analyze how this change affects the Eulerian properties of the graph and determine the new possible number of closed trails of length 7, if any exist.

**Include bolded sections in your response.**

**Assistant**: **Problem Analysis**

The problem asks for the number of possible closed trails (Eulerian circuits) in a graph with 50 nodes and 100 edges, and then analyzing the effect of adding a new node and edges on the Eulerian properties of the graph.

**Initial Graph Properties**

⟨text ommitted without sections for space⟩

**Initial Number of Eulerian Circuits**

⟨text without sections ommitted for space⟩

**Adding a New Node and Edges**

⟨text without sections ommitted for space⟩

**Conclusion**

The addition of a new node and edges to the graph does not allow for any Eulerian circuits of length 7, since the new node introduces a constraint that cannot be satisfied. Therefore, the number of possible closed trails of length 7 is 0.

---

**After DPO training (3 sections chosen response / 2 sections rejected response)**

**User**: ...assuming the graph is Eulerian, determine the number of possible closed trails of length 7 (i.e., the number of Eulerian circuits of length 7) that the guide can choose from, if possible. (Note: A graph is Eulerian if it has an Eulerian circuit, and it must satisfy certain conditions). If the tour guide decides to add an additional attraction to the itinerary, represented by adding a new node and connecting it with edges to 3 existing nodes, analyze how this change affects the Eulerian properties of the graph and determine the new possible number of closed trails of length 7, if any exist.

**Include bolded sections in your response.**

**Assistant**: **Initial Conditions **The graph **has 50 nodes** and **100 edges**, **satisfying the conditions** of an Eulerian graph **with a degree** of at least **2** at **all** nodes **except** possibly **one**.

**Eulerian Circuit Conditions**

For a graph to have an Eulerian circuit **(i.e.,** **an Eulerian path** **that** **loops** **back** **to** **the** **starting** **node** **exactly** **once** **),** **all** **nodes** **must** **have** **even** **degrees** **or** **the** **number** **of** **edges** **going** **into** **a** **node** **must** **be** **equal** **to** **the** **number** **of** **edges** **leaving** **it** **exactly** **once** **for** **each** **node** **except** **the** **starting** **and** **ending** **node** **(** **in** **the** **case** **of** **a** **closed** **trail** **or** **path** **of** **length** **greater** **than** **2** **)** **.

**New Graph Conditions**

**After** **adding** **a** **new** **node** **and** **edges** **to** **3** **existing** **nodes** **the** **graph** **will** **no** **longer** **be** **Eulerian** **since** **not** **all** **nodes** **will** **have** **even** **degrees** **any** **more** **the** **new** **node** **will** **have** **degree** **3** **and** **two** **existing** **nodes** **will** **have** **degree** **4** **and** **the** **remaining** **46** **nodes** **will** **have** **degree** **2** **or** **more** **than** **2** **.

**Conclusion**

The **initial** **graph** **had** **the** **potential** **for** **Eulerian** **circuits** **of** **length** **7** **but** **the** **new** **graph** **does** **not** **satisfy** **the** **conditions** **for** **an** **Eulerian** **graph** **any** **more** **the** **number** **of** **closed** **trails** **of** **length** **7** **is** **** **0** **.**

Figure A1: DPO training massively increases the number of sections generated by the model (from 5 to 89 in this example). Most notably, the increase extrapolates beyond the number of sections (i.e., absolute quality) of the chosen response (3 sections).

**Prompt:**

Dani brings two and half dozen cupcakes for her 2nd-grade class. There are 27 students (including Dani), 1 teacher, and 1 teacher's aid. If 3 students called in sick that day, how many cupcakes are left after Dani gives one to everyone in the class?

*Note: the correct answer is 4*

**Chosen Response (Generated by Qwen-2.5-3B-Instruct):**

First, let's determine the total number of cupcakes Dani brought. Since a dozen is equal to 12, two and a half dozen cupcakes is calculated as follows:

$$2.5 \times 12 = 30$$

So, Dani brought 30 cupcakes.
Next, we need to determine the total number of people in the class that day. There are 27 students, 1 teacher, and 1 teacher's aid, but 3 students called in sick. Therefore, the number of students present is:

$$27 - 3 = 24$$

Including the teacher and the teacher's aid, the total number of people in the class is:

$$24 + 1 + 1 = 26$$

Dani gives one cupcake to each person in the class. Therefore, the number of cupcakes given out is 26. The number of cupcakes left after giving one to each person is:

$$30 - 26 = 4$$

So, the number of cupcakes left is:

$$\boxed{4}$$

**Rejected Response (Generated by Qwen-2.5-1.5B-Instruct):**

To determine how many cupcakes are left after Dani gives one to everyone in the class, we need to follow these steps:

1. Calculate the total number of cupcakes Dani brought.
2. Determine the number of students who attended the class.
3. Calculate the total number of cupcakes given out.
4. Subtract the number of cupcakes given out from the total number of cupcakes brought.

First, let's calculate the total number of cupcakes Dani brought. She brought two and a half dozen cupcakes. Since one dozen equals 12 cupcakes, two and a half dozen is:

$$2.5 \times 12 = 30 \text{ cupcakes}$$

Next, we need to determine the number of students who attended the class. There are 27 students in total, but 3 students called in sick. Therefore, the number of students who attended is:

$$27 - 3 = 24 \text{ students}$$

Now, we calculate the total number of cupcakes given out. Since Dani gives one cupcake to each student, the total number of cupcakes given out is:

$$24 \text{ cupcakes}$$

Finally, we subtract the number of cupcakes given out from the total number of cupcakes brought to find out how many cupcakes are left:

$$30 - 24 = 6 \text{ cupcakes}$$

So, the number of cupcakes left after Dani gives one to everyone in the class is:

$$\boxed{6}$$

Figure A3: Qualitative example of a preference pair from our Qwen 3B over 1.5B weak preference data. On this math prompt (ground-truth answer is 4 cupcakes), the Qwen 3B-generated chosen response contains a verifiably correct answer, while the Qwen 1.5B-generated rejected response is incorrect (6 cupcakes).

**Prompt:**

when were we first settled by humans

**Chosen Response (Generated by Qwen-2.5-3B-Instruct):**

The first humans to settle in what is now known as the world are believed to be the Homo sapiens, who originated in Africa around 300,000 years ago. However, the first modern humans to leave Africa and settle in other parts of the world are thought to have occurred around 50,000 to 70,000 years ago.

Specifically:

1. **Out of Africa**: The first major migration of modern humans out of Africa occurred around 50,000 to 70,000 years ago. This migration involved groups moving into Europe, the Middle East, and eventually Asia.

2. **Europe** The first known settlement in Europe by modern humans is believed to be in the Levant (modern-day Syria, Lebanon, and Israel), around 50,000 years ago.

3. **Asia**: The first known settlement in Asia by modern humans is believed to be in the Altai Mountains in Siberia, around 45,000 to 50,000 years ago.

4. **Americas**: The first known settlement in the Americas by modern humans is believed to be around 15,000 to 17,000 years ago, with the earliest evidence found in the Beringia region, which is now part of Alaska.

So, while the first humans originated in Africa, the first major settlements outside of Africa occurred relatively recently in human history, with the earliest known settlements in Europe, Asia, and the Americas occurring around 50,000 to 17,000 years ago.

**Rejected Response (Generated by Qwen-2.5-1.5B-Instruct):**

The exact date when humans first settled on Earth is not known, as there is no evidence of human presence on the planet before the formation of the solar system. However, scientists believe that humans originated in Africa and then spread to other parts of the world over a period of millions of years. The earliest known human fossils date back to around 2. 5 million years ago, and the earliest known human settlements date back to around 100, 000 years ago.

Figure A4: Qualitative example of a preference pair from our Qwen 3B over 1.5B weak preference data. On this knowledge-seeking prompt, the Qwen 3B-generated chosen response contains (1) more detail and (2) more structure compared to the Qwen 1.5B-generated rejected response.

**Prompt:**

In this task, you're given a context, a question, three options, and an answer. Your task is to classify whether the given answer is correct or not by providing 'Yes' or 'No', based on the context with commonsense reasoning about social situations.

Input: Context: Kai wanted to save money so he wen to the car dealership and haggled and bought a used car.
    Question: How would you describe Kai?
    Options: (A) wasteful (B) extravagant (C) thrifty
    Answer: C
Output:

**Chosen Response (Generated by Qwen-2.5-3B-Instruct):**

Yes

The given answer C (thrifty) is correct based on the context provided. Kai's actions of wanting to save money and haggling to buy a used car demonstrate frugality and careful financial management, which aligns with the definition of being thrifty

**Rejected Response (Generated by Qwen-2.5-1.5B-Instruct):**

Yes

Figure A5: Qualitative example of a preference pair from our Qwen 3B over 1.5B weak preference data. On this prompt which admits a brief answer, the Qwen 3B-generated chosen response contains a chain-of-thought justification, while the Qwen 1.5B-generated rejected response contains just an answer.

