# OpenReview forum: "The Delta Learning Hypothesis: Preference Tuning on Weak Data can Yield Strong Gains"
_colmweb.org/COLM/2025/Conference — COLM 2025_

### Official Review · Reviewer_wem6 · 2025-04-25

**Rating:** 6
**Confidence:** 2
**Ethics Flag:** 1

**Summary:**

The paper proposes the ​​Delta Learning Hypothesis​​, arguing that the relative quality difference (delta) in preference data is more important than absolute quality—even weak paired data can improve model performance through preference tuning (e.g., DPO). Experiments show that training a stronger 8B model with preference data generated by weak models (Qwen-3B) and weaker models (Qwen-1.5B) matches the performance of SOTA methods relying on strong supervision. This approach reduces dependence on strong supervision, offering a low-cost, high-performance alternative for model tuning.

**Questions To Authors:**

1. Can we take the difference between the model itself and a smaller model? Why taking the difference between two smaller models?
2. Do the two models used for differencing need to be from the same series? For example, the paper uses LLama-3B over LLama-1B, or Qwen-3B over Qwen-1.5B. What would happen if we used Llama-3B over Qwen-1.5B?

**Reasons To Accept:**

1. Reduce data cost. This method requires less human labeling.
2. Experiments shows promising results.
3. Paper compares the effectiveness of preference tuning under different model size deltas.

**Reasons To Reject:**

1. The algorithm described in the paper sounds promising, but it remains unclear whether the experimental results on the 8B scale can be generalized to larger models. For instance, in the case of Tulu-405B, the weak models would be the 70B class. Can models of 70B paired with a even weaker model truly teach the 405B model something?
2. In Table 4, the selected weak model Qwen-3B has better average performance than the base model Tulu-8B. Would this affect the conclusion? Is there an upper limit to the capability gap between the weak model and the base SFT model?

---

> ### Author Response · Authors · 2025-06-03
> **Thank you for your valuable feedback! We address your points below (1/2)**
>
> Thank you for your valuable feedback! We are thrilled you found our results promising. Below, we address your thoughtful questions and will revise the paper accordingly.\
> \
> **Concerns:**
>
> &ensp; **1. Results may not generalize to larger models such as Tülu3 405B.**
>
> Thanks for the question! We agree that validating delta learning at larger scales is exciting (`L552-553`). However, such experiments are beyond our academic resource limits, as training a larger-than-8B model requires multiple GPU nodes and specialized infrastructure. For example, training one 405B model takes ~30 hours on 256 interconnected H100s [1], which can cost over $20,000 even before accounting for interconnects [2].
>
> Given these constraints, we prioritized rigor at the widely-deployed 8B scale, which aligns with existing academic best practices [3]. Notably, recent work shows that data findings at 8B often transfer well to larger scales [4]. We train over 200 8B models across different data mixes, carefully tuning hyperparameters to ensure robust conclusions; with our resources, this level of rigor is infeasible at larger scales. We will revise our paper to include this discussion, and to reiterate that delta learning remains to be confirmed at larger scales. Thank you for your feedback!
>
> [1] Tülu 3, Lambert et al. (arXiv 2024)\
> [2] https://lambda.ai/service/gpu-cloud/pricing \
> [3] SimPO, Meng et al. (NeurIPS 2024)\
> [4] Small-to-Large Generalization, Khaddaj et al. (ICLR 2025)\
> \
> &ensp; **2. Qwen-3B outperforms Tülu3-8B-SFT on average—does this affect conclusions? Is there a limit to the weak chosen model – base SFT model gap?**
>
> To clarify a potential miscommunication, our Table 4 experiments test if delta learning can be used to post-train a base SFT model when the supervisors are “near or below” its performance (`L204`). This setup challenges the typical approach of relying on much stronger supervisors like Qwen-72B (`L217-218`), which outperforms the base Tülu SFT model by 14.8 points. In our setup in `Table 4`, Qwen-3B is marginally better but comparable to Tülu on average (+0.3 points) and performs strictly worse on several tasks (e.g., -7.3 IFEval, -29.7 DROP, -11.6 BBH). Yet training Tülu with Qwen-3B supervision still yields state-of-the-art average performance, including gains on the specific tasks where Qwen-3B underperforms (+7.3 IFEval, +0.8 DROP, +0.2 BBH).
>
> Even Qwen-1.5B supervision—11.4 points worse than Tülu—yields consistent gains (`Table 4`, `L237-242`). These results, along with the gains we see in controlled studies using data strictly weaker than the base model (`Section 3.1`), strongly supports our conclusion that delta learning can succeed with weak data. That said, we do suspect a lower limit to how weak the supervision can be. For example, `Figures 3,4` show that gains with Qwen-1.5B supervision are lower than the delta magnitude would predict (`L272-275`, `L302-305`). We speculate that when the chosen model is too weak, the delta to even weaker responses is less informative, as it may be along a dimension that is already obvious to the base model. We leave precise characterization of this limit to future work and will revise our paper with this discussion. Thank you!\
> \
> **Questions:**
>
> &ensp; **1. Can we take the difference between the model itself and a smaller model?**
>
> Yes! `Section 3.2` explores this in detail; we show that training Llama-8B to prefer its own greedy generations over those from Llama-3B yields consistent gains. The gains are relatively modest. To conjecture, this may be because the model already assigns high likelihood to self-generations; the model already “knows” to prefer its own output, and reinforcing the delta adds less new information. We leave a deeper understanding of the optimization dynamics with self-generated data to future work, and will revise to discuss these interesting open questions.

---

> > ### Author Response · Authors · 2025-06-03
> > **Thank you for your valuable feedback! We address your points below (2/2)**
> >
> > **Questions:**
> >
> > &ensp; **2. What if we pair models from two families, such as Llama 3B over Qwen 1.5B?**
> >
> > Great question! Our original motivation for using a weak and weaker model from the same series was that they are trained with largely similar data and architecture and differ primarily in parameter count. This setup increases the likelihood that the larger model is consistently stronger, thereby reducing the risk of generating noisy preference pairs where the smaller model may produce better outputs than the larger one.
> >
> > That said, nothing fundamentally prevents us from pairing models across families. Motivated by your question, we ran additional experiments using cross-family model pairs:  (1) Llama-3B over Qwen-1.5B (as you suggested), (2) Llama-3B over Qwen-0.5B, and (3) Qwen-3B over Llama-1B. We follow the hyperparameters of our analysis experiments (`Appendix D.6`). The results in `Table R4` below show that cross-family pairings yield consistent improvements in average performance. However, the gains are generally smaller than with same-family pairings. We speculate this is due to increased noise in the preference pairs: for instance, on MATH, Qwen-1.5B outperforms Llama-3B, so training with Llama-3B over Qwen-1.5B introduces a delta in the wrong direction and subsequently hurts MATH performance after tuning. We will add these results to our paper. Thank you for the suggestion!\
> > \
> > **Table R4: Results of training Tülu-3-8B-SFT on preference data from cross-family model pairs**
> > | Model Name | MMLU | PopQA | MATH | GSM  | AE2  | IFEval | BBH  | DROP | TQA  | HEval | HEval+ | Avg. |
> > | ------- | ---- | ----- | ---- | ---- | ---- | ------ | ---- | ---- | ---- | ----- | ------ | ---- |
> > | Llama-3.2-1B-Instruct | 46.1 | 13.9 | 21.1 | 44.4 | 8.8 | 54.5 | 40.2 | 32.2 | 40.0 | 64.8 | 60.0 | 38.7 |
> > | Llama-3.2-3B-Instruct | 62.9 | 19.4 | 39.6 | 75.7 | 18.7 | 76.5 | 61.6 | 48.5 | 50.6 | 79.7 | 76.8 | 55.5 |
> > | Qwen-2.5-0.5B-Instruct | 46.2 | 10.1 | 27.2 | 39.2 | 3.3 | 28.8 | 32.2 | 25.3 | 45.4 | 60.5 | 58.9 | 34.3 |
> > | Qwen-2.5-1.5B-Instruct | 59.7 | 15.4 | 41.6 | 66.2 | 7.2 | 44.2 | 45.9 | 14.1 | 46.5 | 83.0 | 79.8 | 45.8 |
> > | Qwen-2.5-3B-Instruct | 69.5 | 15.7 | 63.1 | 77.7 | 17.8 | 64.0 | 57.6 | 31.5 | 57.2 | 90.5 | 87.4 | 57.5 |
> > | **―――――――――――** |
> > | **Tülu-3-8B-SFT** | 66.1 | 29.6 | 31.2 | 76.0 | 12.2 | 71.3   | 69.2 | 61.2 | 46.8 | 86.2  | 79.8   | 57.2 |
> > | + Llama-3B over Qwen-1.5B | 66.4 | 29.5  | 27.4 | 76.5 | 16.8 | 73.0   | 68.3 | 60.9 | 50.6 | 87.4  | 81.7   | 58.0 |
> > | + Llama-3B over Qwen-0.5B | 67.0 | 29.7  | 31.2 | 78.1 | 18.3 | 75.2   | 69.1 | 63.3 | 54.2 | 84.7  | 79.3   | 59.1 |
> > | + Qwen-3B over Llama-1B | 69.3 | 30.7  | 33.8 | 83.8 | 26.4 | 73.9   | 68.1 | 56.3 | 57.0 | 85.1  | 80.3   | 60.4 |
> > | **―――――――――――** |
> > | + Tülu 3 Preference Dataset | 69.8 | 30.3  | 42.6 | 84.2 | 32.8 | 80.4   | 69.2 | 62.5 | 56.1 | 84.7  | 80.8   | 63.0 |
> >
> > \
> >  **――――――――――――――――――――――――**
> >
> > Thank you again for taking the time to review our work. We look forward to further discussion!

---

> > > ### Comment · Reviewer_wem6 · 2025-06-03
> > >
> > > Thank you for your detailed response.  I understand that the training cost for larger-scale models is very high and indeed beyond the scope of academia. From the perspective of the paper's soundness, I still have some reservations about the effectiveness of larger-scale models. My other concerns have been addressed, so I maintain my original score.

---

> > ### Author Response · Authors · 2025-06-06
> > **Thank you for your continued engagement! As your concerns have been addressed, we hope you'll consider increasing your score.**
> >
> > Thank you again for your thoughtful review! We truly appreciate your time and engagement, and we’re thrilled to hear that all but one of your concerns—scaling to 405B—have been addressed.
> >
> > We agree that validating delta learning at frontier scales would be exciting future work. However, we would like to reiterate that our results at the 8B scale are robust, reproducible, and clearly demonstrate the promise of delta learning. As you noted, like most academic researchers, we do not have access to the compute required to train 405B models. We hope you’ll consider these constraints in light of COLM’s reviewing guidelines, which recognize that meaningful innovation can come without experiments at the “upper limit of scale,” and that “limiting” research to only those with massive resources may stifle progress [1]. While not all 8B-scale results may hold at frontier scales, "some results will." We are optimistic that delta learning falls into this category.
> >
> > Given that your other concerns have been addressed, we sincerely hope you’ll consider increasing your score. We look forward to further discussing with you—thank you for your time!
> >
> >
> > [1] https://colmweb.org/ReviewGuide.html

---

> > ### Author Response · Authors · 2025-06-09
> > **Thank you for your time! We eagerly await your response for further discussion.**
> >
> > Thank you again for your feedback! Based upon your review, we plan to revise our paper by moving our discussion on scale limitations (`Appendix B, L551-552`) to our main text. We also plan to revise the discussion to include the following text:
> >
> > > In this work, we have shown that models can learn surprisingly well from the delta between paired weak data points. At the widely-deployed 8B scale, we have used this insight to greatly simplify open post-training recipes. We are optimistic that our work may open future research on what other deltas---beyond the overall model quality deltas explored in this paper---may be informative, and what tasks such deltas might benefit. Moreover, our work focuses on the 8B scale; it remains an exciting open question to validate delta learning at larger frontier scales, and to understand how the properties of what makes a delta useful might change as models get larger. We leave these questions to future work.
> >
> > We would love to hear your thoughts, as well as any further feedback you may have for strengthening our paper. We eagerly await your response to our comments!

---

### Official Review · Reviewer_Mjit · 2025-05-01

**Rating:** 9
**Confidence:** 3
**Ethics Flag:** 1

**Summary:**

This paper introduces the *delta learning hypothesis*, arguing that DPO using paired weak data, where both chosen and rejected responses are generated by models weaker than the one being trained, can yield performance improvements beyond directly SFT the model on the weak data. The authors validate this hypothesis using a toy setting and present a simple and cost-effective pipeline for preference tuning. They demonstrate that models can benefit significantly from learning the *relative quality difference* ("delta") between responses, even when the absolute quality of each response is low.

The paper is well written and easy to follow. The proposed idea is efficient and effective. The ablation study also supports the claims well. Overall, this is a high-quality work and might have a broader impact on the CoLM’s community.

**Questions To Authors:**

- As mentioned by the authors, we might not expect that all responses generated by weaker models are worse than the weak model. Hence, the scope of this approach might not be so general as expected. However, as shown by the experiments, the model can still learn useful information when the delta is noisy.
- Maybe showing how weak responses are better than weaker responses via some examples would be helpful.
- Maybe more discussions about what scope the proposed methods would bring more benefits (e.g., reasoning, truthful QA, general alignment, etc.) would be helpful.

**Reasons To Accept:**

- Novel and impactful hypothesis: Rather than SFT the model uses the weak responses, creating supervisory signals by comparing weak and weaker responses is a more natural way to conduct weak-to-strong learning.
- Extensive empirical verifications: The paper contains experiments on different scales and datasets. The demonstration of a toy example is also inspiring.
- Cost efficiency: The proposed method is simple and effective, reducing the huge cost of using strong models to generate responses.

**Reasons To Reject:**

N/A

---

> ### Author Response · Authors · 2025-06-03
> **Thank you for your valuable feedback! We address your points below.**
>
> Thank you for your valuable review! We are thrilled you found our hypothesis novel and impactful, and our research useful for the COLM community. We answer your questions below, and hope our responses lend deeper insight into our work.\
> \
> **Questions:**
>
> &ensp; **1. Not all responses generated by the weaker model may be worse than the weak model.**
>
> This is a great observation! As you noted, we empirically found that learning with weak preference data still succeeds even with potentially noisy deltas (`L210-212`, `Table 4`). That said, we agree that training with clean deltas—where all rejected responses are strictly worse than the chosen ones—could potentially yield more robust performance. Therefore, we believe it would be valuable to (a) study the performance difference between delta learning with noisy versus clean deltas, and (b) explore new methods for curating paired data with cleaner deltas. One such approach could involve synthetically generating rejected responses through targeted corruptions of existing responses, ensuring that the rejected responses are reliably worse. In our revised paper, we will re-emphasize these points to better highlight these exciting avenues for future research. Thank you for your valuable feedback!\
> \
> &ensp; **2. It would be useful to see qualitative examples of how weak responses are better than weaker responses.**
>
> Thank you for the valuable suggestion! To better understand the delta between weak and weaker responses, we manually inspected pairs from our best post-training preference data (Qwen-3B over Qwen-1.5B). Overall, there were no universal axes along which the Qwen-3B responses were better than the Qwen-1.5B responses, but we did observe several interesting deltas. We summarize them here, and showcase (abbreviated) qualitative examples of these differences in the next comment below.
>
> - On prompts with verifiable answers (e.g. math, code), we find pairs where Qwen-3B responds correctly but Qwen-1.5B does not (`Example 1` below)
> - On knowledge-seeking prompts, we find pairs where Qwen-3B responds with more detail (`Example 2` below)
> - On prompts that admit brief answers, we find pairs where Qwen-3B generates a chain-of-thought, while Qwen-1.5B responds with just the answer (`Example 3` below)
>
> We believe it would be exciting for future work to further characterize how these observed deltas in the preference pairs translate to downstream model behavior after training. Thank you again for suggesting this analysis! We will include this discussion in the next version of our paper.\
> \
> &ensp; **3. What capabilities do the proposed methods benefit most?**
>
> Good question! Empirically, with our best delta learning post-training setup (i.e., Qwen-3B over Qwen-1.5B pairs), we observe the strongest gains over the base SFT model on general instruction following (+15.6 points; AlpacaEval2, IFEval), mathematical reasoning (+9.4; MATH, GSM), and truthfulness (+10.9; TruthfulQA). We observe consistent but more modest gains on knowledge recall tasks (+2.7; MMLU, PopQA), coding (+3.5; HumanEval, HumanEval+), and general reasoning (+0.5; BigBenchHard, DROP). These gains are computed from our results in `Table 4`. Interestingly, the original Tülu 3 preference data yields a similar pattern of improvements. Thus, it remains an exciting open question as to whether (a) delta learning is particularly effective for enhancing math, instruction following, and truthfulness, or (b) the prompts in the Tülu 3 dataset—which our data shares—are especially conducive to improvements in these capabilities, and how we might curate prompts to improve gains in the other capabilities.
>
> Finally, we are optimistic that our controlled experiments on bolded section generation—where we constructed paired data with deltas along a specific semantic axis—provide early evidence that delta learning can drive targeted improvements in the corresponding capability. This suggests that it may be possible to tailor deltas to shape specific behaviors in the model. We leave further study of such targeted deltas to future work. We will revise our paper to include this discussion, thank you!\
> \
> \
> We look forward to further discussions!

---

> > ### Author Response · Authors · 2025-06-03
> > **Additional qualitative examples**
> >
> > **EXAMPLE 1:**
> >
> > **Prompt:** Dani brings two and half dozen cupcakes for her 2nd-grade class.  There are 27 students (including Dani), 1 teacher, and 1 teacher’s aid.  If 3 students called in sick that day, how many cupcakes are left after Dani gives one to everyone in the class?
> >
> > **Chosen response (Qwen-2.5-3B-Instruct)**:
> > > First, let's determine the total number of cupcakes Dani brought. Since a dozen is equal to 12, two and a half dozen cupcakes is calculated as follows…\
> > > \
> > > So, the number of cupcakes left is:
> > >
> > > \[ \boxed{4} \]
> >
> >
> > **Rejected response (Qwen-2.5-1.5B-Instruct)**:
> > > To determine how many cupcakes are left after Dani gives one to everyone in the class, we need to follow these steps…\
> > > \
> > > So, the number of cupcakes left after Dani gives one to everyone in the class is:
> > > \[ \boxed{6} \]
> >
> > Note: the correct answer is 4.
> >
> > \
> >  **――――――――――――――――――――――――**
> >
> > **EXAMPLE 2:**
> >
> > **Prompt:** when were we first settled by humans
> >
> > **Chosen response (Qwen-2.5-3B-Instruct)**:
> > > The first humans to settle in what is now known as the world are believed to be the Homo sapiens, who originated in Africa around 300,000 years ago. However, the first modern humans to leave Africa and settle in other parts of the world are thought to have occurred around 50,000 to 70,000 years ago.\
> > > \
> > > Specifically:
> > >
> > > 1. **Out of Africa**: The first major migration of modern humans out of Africa occurred around 50,000 to 70,000 years ago. This migration involved groups moving into Europe, the Middle East, and eventually Asia.
> > > 2. **Europe**: The first known settlement in Europe by modern humans is believed to be in the Levant (modern-day Syria, Lebanon, and Israel), around 50,000 years ago.
> > > 3. **Asia**: The first known settlement in Asia by modern humans is believed to be in the Altai Mountains in Siberia, around 45,000 to 50,000 years ago.
> > > 4. **Americas**: The first known settlement in the Americas by modern humans is believed to be around 15,000 to 17,000 years ago, with the earliest evidence found in the Beringia region, which is now part of Alaska.
> > >
> > > So, while the first humans originated in Africa, the first major settlements outside of Africa occurred relatively recently in human history, with the earliest known settlements in Europe, Asia, and the Americas occurring around 50,000 to 17,000 years ago.
> >
> > **Rejected response (Qwen-2.5-1.5B-Instruct)**:
> > > The exact date when humans first settled on Earth is not known, as there is no evidence of human presence on the planet before the formation of the solar system. However, scientists believe that humans originated in Africa and then spread to other parts of the world over a period of millions of years. The earliest known human fossils date back to around 2. 5 million years ago, and the earliest known human settlements date back to around 100, 000 years ago.
> >
> > \
> >  **――――――――――――――――――――――――**
> >
> > **EXAMPLE 3:**
> >
> > **Prompt**:
> > Classify whether the given answer is correct or not…
> > …Kai wanted to save money so he went to the car dealership and haggled…
> >  Question: How would you describe Kai?
> >  Options: (A) wasteful (B) extravagant (C) thrifty
> >  Answer: C
> > Output:
> >
> > **Chosen response (Qwen-2.5-3B-Instruct)**:
> >
> > > The given answer C (thrifty) is correct based on the context provided. Kai's actions of wanting to save money and haggling to buy a used car demonstrate frugality and careful financial management, which aligns with the definition of being thrifty.
> >
> > **Rejected response (Qwen-2.5-1.5B-Instruct)**:
> > > Yes

---

> > > ### Comment · Reviewer_Mjit · 2025-06-06
> > >
> > > Thanks for the author's response, which is very insightful and might further strengthen the influence of this paper. Curating paired data with cleaner deltas would also be a valuable future direction based on the findings in this paper, which might play an important role in weak-to-strong style self-improving methods. I hence increase my evaluation to 9!

---

> > > > ### Author Response · Authors · 2025-06-06
> > > > **Thank you for your time!**
> > > >
> > > > Thank you for your reply! We're thrilled that our response addressed your questions, and appreciate your score increase. We will revise our paper according to the valuable feedback you've provided. Thank you again for helping us strengthen our work!

---

### Official Review · Reviewer_vyR3 · 2025-05-09

**Rating:** 9
**Confidence:** 5
**Ethics Flag:** 1

**Summary:**

The paper validates an interesting hypothesis: can a preference dataset created with weaker models improve the performance of a better (e.g., larger) language model? The authors demonstrate that this is indeed possible as long as there is a clear quality delta between the preferences (i.e., the chosen sample needs to be clearly better than the rejected sample). First, the authors carry out a pilot study leveraging preferences from the UltraFeedback dataset, keeping only preferences generated by models that are weaker or equivalent to a Llama 3B model. They show that tuning a Llama 8B and a Llama 3B on that datasets can lead to performance clear improvements. Then, the authors carry out two controlled studies. The first one generates a toy dataset where the model is trained to prefer generation with more bolded headings. They show that the model can learn to generate outputs with substantially higher number of headings, although the preference dataset only contains 3 headings in the chosen sample and 2 headings in the rejected sample. Instead, training the model in a supervised fine-tuned manner over the chosen samples does not make the model generate more headings. In the second controlled study, they generate a preference dataset using a Llama 8B for the chosen samples and Llama 3 B for the rejected samples, and show a Llama 8B model can improve from using this data. Finally, the paper explores their hypothesis against a state-of-the-art preference tuning data recipe, which is used to generate the Tulu 8B DPO model, and leverages preferences generated from a diverse set of LLMs, which are labelled using a frontier model such as GPT-4o. The experiments demonstrate that a simpler and cheaper data recipe leveraging different smaller LLMs from the Qwen 2.5 family, can generate a DPO preference set that can yeld a better improved Tulu 8B model. Additional sensitivity and ablation analysis sheds more light into the delta learning hypothesis.

**Reasons To Accept:**

- The paper is very well written and the research is well motivated.
- Strong experimental results over a range of datasets and language models of different sizes, which answer the main research question, a preference dataset generated from weaker models can improve a stronger LLM baseline.
- Interesting sensitivity analysis, showing that performance improvements plateau after a certain point of a preference delta.
- The paper presents a valuable finding. Other researchers and LLM practitioners can leverage this way of generating preference datasets from weaker models in various tasks.

**Reasons To Reject:**

- Missing details on the generated preference data. It would be interesting to show some statistics on the different preferences generated by the different Qwen models (e.g., average length, vocabulary diversity, similarity across different models).
- Missing information regarding the original preference dataset used for the Tulu 8B-DPO model. The details of that dataset may be published in their original paper, but it would be good to present them here too. Is the number of preferences in that dataset comparable to the one generated by the authors using the Qwen 2.5 models?
- Some preliminary results such as the controlled study for the number of sections generated and a similar controlled study with the Llama 8B and the Llama 3B models (using a fewer number of datasets) have been previously published.

---

> ### Author Response · Authors · 2025-06-03
> **Thank you for your valuable feedback! We address your points below (1/2)**
>
> Thank you for your valuable review! We are glad you found our work well-motivated and valuable to both researchers and practitioners. We address your concerns below and hope our responses lend more insight into our work.
>
>
>
> **Concerns:**
>
> &ensp; **1. Missing details in the delta learning preference data; it would be interesting to show more statistics.**
>
> Thank you for the excellent suggestion! To address your concern, we computed the following statistics for each dataset used in our core post-training experiments (`Table 4`), using the Tülu-3-8B-SFT tokenizer (i.e. the base SFT model’s tokenizer):
>
> - _Average token length_ of chosen and rejected responses
> - _Vocabulary diversity_, measured as the number of unique 1-gram and 2-gram tokens in chosen and rejected responses
> - _Cosine similarity_ between chosen and rejected response embeddings, computed using OpenAI's `text-embedding-3-small` API
>
> As a reference point, we also report these statistics for the original Tülu 3 dataset. Results are shown in `Table R3` (next comment below).
>
> Overall, we observe that the chosen responses in our weak preference data are generally (a) shorter and (b) more diverse in vocabulary than the paired rejected responses. In contrast, chosen and rejected responses in Tülu 3 exhibit largely similar statistics. We hypothesize that the longer length of rejected responses in our data may be due to degenerate outputs from weak models; they sometimes repeat tokens until reaching the maximum generation length. The cosine similarity between chosen and rejected responses is relatively small for all datasets; qualitatively, chosen and rejected responses often differ significantly on a surface semantic level, which may contribute to this low embedding similarity.
> We will include all these details in the updated version of our paper. Thank you!\
> \
> &ensp; **2. Missing details on how the original Tülu 3 dataset was created. Is it comparable in size to the datasets generated in this paper?**
>
> Thank you for the question! We will revise our Appendix to include a complete description of how the original Tülu 3 preference data was constructed (i.e., details of prompt distribution, what models were used for generations, GPT-4o judge details). We will also revise our description of Tülu 3 in the main text (`Section 4.1`) to refer interested readers to these additional details.
>
> All of our weak preference datasets (including the ones generated with Qwen) were constructed to be the same size and with the same prompts as the original Tülu 3 dataset, minus the \~6000 duplicate prompts in the Tülu 3 data (`Appendix D.5`). In other words, our data is constructed from the 264,806 unique prompts in the Tülu 3 dataset, which contains \~271k prompts total (\~6k duplicates). We will revise our paper to make this more clear in the main text, thank you!\
> \
> &ensp; **3. Preliminary versions of the controlled studies in this paper (bold sections, Llama 8B self-improvement) have been previously published.**
>
> Thank you for the feedback! Could you please point us to the relevant works? We are not aware of such studies, but would be more than happy to cite them and discuss with you to contextualize them against our work.
>
> To our understanding, the closest related works are [1] Varying Shades of Wrong (Yao et al. 2024) and [2] Weak-to-strong Preference Optimization (Wang et al. 2025) (`L34-35`). [1] shows that training on preference pairs where the chosen and rejected responses are both verifiably wrong—but the chosen response is less wrong (i.e. closer to correct answer by some metric)—can lead to gains on tasks such as Knowledge Crosswords and biography generation. They focus on comparing various methods for labeling "wrong-over-wrong" preference pairs, and find that using a GPT-4 judge-based method performs best. [2] proposes a modified DPO objective to align a larger unaligned model (i.e. 7B) using the distributional differences of a smaller model (i.e. 1.5B) before and after alignment. Motivated by these studies offering preliminary evidence that preference tuning can enable models to surpass the quality of their supervision, we formalize the delta learning hypothesis, present fully controlled experiments to validate it, and finally show its efficacy for state-of-the-art post-training. We will revise our related works section to include this expanded discussion. Thank you for your feedback!\
> \
> \
> We look forward to further discussion!

---

> > ### Author Response · Authors · 2025-06-03
> > **Thank you for your valuable feedback! We address your points below (2/2)**
> >
> > **Table R3: Statistics of the chosen and rejected responses in our weak preference datasets, with statistics of responses from the Tülu 3 preference dataset shown for reference**
> >
> > | Chosen Response            | Rejected Response          | Chosen Avg. Length | Rejected Avg. Length | Chosen Uniq. 1-Grams | Rejected Uniq. 1-Grams | Chosen Uniq. 2-Grams | Rejected Uniq. 2-Grams | Chosen-Rejected Cosine Sim. |
> > |-------------|--------|--------|-----------------------|-------------------------|---------------------------|------------------------|--------------------------|-------------------|
> > | Qwen-2.5-3B-Inst.    | Qwen-2.5-1.5B-Inst.  | 709.1               | 776.5                 | 114,183                 | 110,380                   | 6,637,312              | 5,107,761                | 0.115             |
> > | Llama-3.2-3B-Inst.   | Llama-3.2-1B-Inst.   | 779.3               | 1041.3                | 112,553                 | 109,090                   | 6,858,616              | 5,884,941                | 0.117             |
> > | Qwen-2.5-1.5B-Inst.  | Qwen-2.5-0.5B-Inst. | 776.5               | 1317.8                | 110,380                 | 106,900                   | 5,107,761              | 4,626,119                | 0.112             |
> > | **―――――――――** |
> > | Tülu 3 Chosen           | Tülu 3 Rejected         | 443.9               | 441.3                 | 119,269                 | 119,320                   | 10,980,913             | 10,866,352               | 0.117             |

---

> > ### Comment · Reviewer_vyR3 · 2025-06-07
> >
> > Thank you for a detailed response to my concerns.
> >
> > Regarding my last comment (on the preliminary versions of this paper), I came across a preliminary version of this work in a workshop presentation. When I wrote my review, I had not noticed that the workshop was a non-archival venue.
> >
> > Overall, I think this is a great work. I will raise my score to 9.

---

> > > ### Author Response · Authors · 2025-06-07
> > > **Thank you for your time!**
> > >
> > > Thank you for your reply! We're thrilled that you enjoyed our work and appreciate your score increase. We will revise our paper according to the valuable feedback you've provided in review. Thank you again for helping us strengthen our work!

---

### Official Review · Reviewer_nk1W · 2025-05-25

**Rating:** 8
**Confidence:** 5
**Ethics Flag:** 1

**Summary:**

This paper introduces the "delta learning hypothesis", demonstrating that preference tuning using weak supervision (even when both examples are individually “weak” relative to the model’s capabilities) can drive substantial performance gains via preference tuning. The authors empirically demonstrate this in controlled experiments (e.g., synthetic bold-header generation), semi-controlled setups (e.g., preferring 8B model outputs over 3B ones), and large-scale post-training (e.g., tuning Tulu-3-8B using responses from Qwen 2.5 3B vs. 1.5B).

The authors show that tuning with preference pairs based on weak models can match or even exceed the performance of existing state-of-the-art pipelines like Tulu 3, which rely on strong supervision from GPT-4o and other large models. They explore key dimensions of the hypothesis, such as the magnitude of the delta, the absolute chosen quality, and the generalizability across base models (e.g., OLMo). The work is well-executed and positions delta-based learning as a compellingly simple and cheap alternative to high-resource preference tuning.

**Questions To Authors:**

1. How does delta learning perform on tasks outside the 11‐benchmark suite, particularly safety‐critical evaluations? Are there tasks or datasets where delta learning fails to improve or actively harms model performance?
2. If one has limited access to expert data or strong instruction datasets, would delta learning from weak relative signals be a more scalable or reliable strategy than collecting absolute SFT examples? In other words, does delta learning act as a complement to instruction tuning or can substitute for it in some regimes?
3. What is the performance degradation (if any) when the delta is incorrect (i.e., when the smaller model's output is actually better)?
4. Tuning with weak data might amplify undesirable patterns in weak responses, so how do you ensure that preference tuning avoids learning from spurious or poor-quality responses (maybe) present in weaker models?

**Reasons To Accept:**

1. The paper clearly formulates delta learning as a novel hypothesis, challenging the assumption that absolute data quality must exceed model capability.
2. Provides strong empirical results across baselines showing consistent performance gains using only weak models for both response generation and preference labelling.
3. The authors have laid out well-defined ablations, i.e, the relative quality delta, absolute quality magnitude, saturation effects and other heuristic validity, providing ample insights into when and why delta learning works.
3. Overall, the proposed recipe is more accessible, much cheaper and eliminates the reliance on GPT-4o while retaining SOTA performance.
4. The paper is well-written, well-described and has extensive experimental details, and clear metrics. Also provides detailed appendices on training, hyperparameters.

**Reasons To Reject:**

1. The gains are mostly reported on standardised LLM benchmarks. It remains unclear how delta learning performs in safety benchmarks, or bias assessments or domain-specific downstream tasks.
2. Though I enjoyed reading the paper, and it presents strong empirical evidence for delta learning, the paper can stand out with a discussion on the mechanistic explanation of why delta learning works, beyond high-level intuition.
3. While delta magnitude is studied, the paper leaves some gaps and open questions on what semantic properties (beyond size proxies) make a delta most informative.

---

> ### Author Response · Authors · 2025-06-03
> **Thank you for your valuable feedback! We address your points below (1/3)**
>
> Thank you for your valuable review! We are glad you found our hypothesis novel and our experiments compelling. We will revise our paper based on your suggestions. In particular, we are excited to share new theoretical results proving that delta learning succeeds for logistic regression, and hope the proof sheds further insight on why delta learning works. We now address each of your points below:\
> \
> **Concerns:**
>
> &ensp; **1. Gains are shown on standard benchmarks; how does delta learning perform for other behaviors like safety?**
>
> Thanks for the valuable suggestion! We agree that safety is critical to study (`L550`). Our submission focused on core capabilities to highlight the promise of delta learning; motivated by your feedback, we evaluated our main post-trained models (`Table 4`) on (a) 3 benchmarks measuring if models refuse unsafe requests (XSTest, HarmBench, and WildGuardTest) and (b) 3 benchmarks measuring jailbreaking vulnerability (Do-Anything-Now, JailbreakTrigger, WildJailbreakTest).
>
> We show results in Table R1 below. Overall, both strongly-supervised Tülu 3 preference data and delta learning tend to slightly hurt average safety compared to the base SFT model. Thus, we conjecture that these drops are due to characteristics of the Tülu 3 prompt distribution [1] shared between these data rather than an inherent limitation of delta learning. Intuitively, we speculate that if the prompts do not expose useful deltas, then one cannot expect gains. Nonetheless, models trained with delta learning degrade _less_, and hence are generally more safe than Tülu-3-8B-DPO. An exception is the model trained with Qwen 3B over 1.5B. While it correctly refuses more often than Tülu-3-8B-DPO, it is easier to jailbreak. We conjecture that this is because Qwen 3B itself is easier to jailbreak than Qwen 1.5B (`Table R1`; see also [2,3] which suggest that model size can sometimes inversely correlate with safety), and hence the delta between Qwen 3B and 1.5B is in a negative direction. How to curate prompts and deltas that effectively improve safety remains an exciting open question; we will revise to include these results and discussions.
>
> [1] Tülu 3, Lambert et al. (arXiv 2024)\
> [2] OLMo 2, Team OLMo (arXiv 2024)\
> [3] Pythia, Biderman et al. (ICML 2023)\
> \
> **Table R1: Safety evaluation of our post-trained models**
>
> | Model Name  | XSTest | HarmB. | WildGuard | DAN  | JailbreakT. | WildJail. | Avg. Refusal ↑ | Avg Jailbreak ↑ | Avg. Overall ↑ |
> |---|--------|-----------|-----------|------|--------------|----------------|----|------|-------------------|
> | Llama-3.2-1B-Instruct     | 81.1   | 65.0      | 78.1      | 87.0 | 74.5         | 61.8           | 74.7           | 74.4             | 74.6        |
> | Llama-3.2-3B-Instruct     | 90.9   | 77.8      | 88.1      | 95.0 | 78.0         | 68.4           | 85.6           | 80.5             | 83.0        |
> | Qwen-2.5-0.5B-Instruct    | 72.2   | 70.3      | 72.5      | 64.7 | 84.8         | 50.9           | 71.7           | 66.8             | 69.2        |
> | Qwen-2.5-1.5B-Instruct    | 71.8   | 94.7      | 79.8      | 77.7 | 82.0         | 53.6           | 82.1           | 71.1             | 76.6        |
> | Qwen-2.5-3B-Instruct      | 87.8   | 91.2      | 83.2      | 49.0 | 67.8         | 56.0           | 87.4           | 57.6             | 72.5        |
> | **―――――――――――** |
> | **Tülu-3-8B-SFT**         | 90.7   | 98.8      | 99.2      | 87.7 | 96.0         | 86.6           | 96.2           | 90.1             | 93.2        |
> | + Llama 3B over 1B        | 93.1   | 97.2      | 99.2      | 73.0 | 88.0         | 85.1           | 96.5           | 82.0             | 89.3        |
> | + Qwen 1.5B over 0.5B     | 90.9   | 98.1      | 99.3      | 84.7 | 94.8         | 88.0           | 96.1           | 89.2             | 92.6        |
> | + Qwen 3B over 1.5B       | 93.6   | 96.2      | 99.1      | 62.3 | 83.0         | 78.8           | 96.3           | 74.7             | 85.5        |
> | **―――――――――――** |
> | + Tülu 3 Preference Data       | 92.9   | 95.3      | 98.5      | 68.7 | 87.2         | 81.3           | 95.6           | 79.1             | 87.3        |

---

> > ### Author Response · Authors · 2025-06-03
> > **Thank you for your valuable feedback! We address your points below (2/3)**
> >
> > **Concerns:**
> >
> > &ensp; **2. The paper has strong empirical evidence, but could stand out with mechanistic understanding of why delta learning works.**
> >
> > Spurred by your feedback, we studied delta learning in a controlled theoretical setting. We prove that delta learning succeeds with high probability for _homogeneous_ logistic regression (homogeneous meaning no bias, so that the decision boundary passes through the origin). We believe the proof offers insight into why delta learning can work, and we sketch key ideas in the problem setup and proof here. We will update our paper to include the longer formal derivation. If you have interest, we would also be more than happy to discuss further details through OpenReview during discussion.
> >
> > **Problem setup.** We assume inputs $x \in \mathbb{R}^d$ are drawn from an isotropic Gaussian with mean 0 and identity covariance, and that there exists some (unobserved) ground-truth model with weights $\theta^\*$ achieving perfect classification accuracy. Fix an arbitrary student model $\theta_0$ that we aim to improve.
> > We construct preference pairs $(x, y_c, y_r)$, where $y_c, y_r$ are chosen and rejected pseudo-labels (respectively) generated by teacher models $\theta_c, \theta_r$. The teacher models are randomly sampled from the set of all logistic regression models, under the constraint that their cosine similarities with the ideal model $\theta^\*$ are some arbitrary but pre-determined constants $-1 < \alpha_r < \alpha_c < 1$:
> > $$
> > \cos(\theta_c, \theta^\*) = \alpha_c,  \cos(\theta_r, \theta^\*) = \alpha_r.
> > $$
> > In other words, we pick a random $\theta_c$ satisfying $\cos(\theta_c, \theta^\*) = \alpha_c$, and likewise for $\theta_r$. Intuitively, higher cosine similarity with $\theta^*$ implies better average performance over the data, so $\alpha_c > \alpha_r$ implies that $\theta_c$ is a stronger model than $\theta_r$ in expectation.
> >
> > We train the student model with SGD to prefer the stronger teacher’s labels $y_c$ over the weaker teacher’s labels $y_r$ using the simple loss,
> > $\mathcal{L} (x,y_c,y_r; \theta) = p_\theta (y_c | x) - p(y_r | x)$. For simplicity, we assume training with the population gradient here; we have also shown an extension to finite empirical gradients via a standard martingale analysis and will present the full results in our final paper.\
> > \
> > **Theorem (informal).** Fix a failure probability $\delta \in (0,1)$. So long as the dimension $d$ of the model exceeds a certain (reasonable) threshold, then with high probability at least $1-\delta$ (over the sampling of teacher models) the training procedure improves the student, even when both teachers are weaker than the student itself. In particular, the threshold depends on the student model's initial alignment with $\theta^\*$ (denoted $\alpha_0$), the size of the strength gap $(\alpha_c -\alpha_r)$ between the teacher models, and the failure probability $\delta$:
> > $$
> >       d > 2\ln \frac{4}{\delta}
> >       \Biggl(
> >          \frac{2|\alpha_{0}|\cdot |\|\theta_{0}\|\| }
> >               {(\alpha_{c}-\alpha_{r}) (1-\alpha_{0}^{2})}
> >       \Biggr)^{2}+1.
> > $$
> > For realistic choices of these parameters, $d \approx 2000$ suffices, i.e. the threshold is not prohibitively high.
> >
> > Our result says that for any arbitrary student model, almost all pairs of teacher models exhibiting a performance delta suffice to generate preference data that will improve it. As a caveat, we have no guarantees on the _magnitude_ of the improvement. In particular, our result requires early stopping; it does not imply that we can keep training the student to get arbitrarily large gains.\
> > \
> > **Proof (informal).** The key idea is that by Stein’s Lemma, the gradient update induced by the loss moves the student in the direction of $(\theta_c - \theta_r)$—towards the stronger teacher $\theta_c$ and away from the weaker teacher $\theta_r$. Since $\theta_c$ is, by assumption, better aligned with the optimal model $\theta^\*$ than $\theta_r$, the difference vector $\theta_c - \theta_r$ is itself positively aligned with $\theta^\*$, regardless of how low the absolute alignment of $\theta_c$ may be. **In other words, the delta between the two teachers yields a directionally correct signal, even when both teachers are individually weak.** In high dimensions, the noise components of the update become negligible and this directional signal dominates. Thus, with high probability, training improves the student, even when the student is already better than either teacher.\
> > \
> > We will discuss these results in our updated paper. Thank you for helping strengthen our work!

---

> > ### Author Response · Authors · 2025-06-03
> > **Thank you for your valuable feedback! We address your points below (3/3)**
> >
> > **Concerns:**
> >
> > &ensp; **3. What semantic properties of the delta (beyond magnitude) matter remains an open question.**
> >
> > This is an exciting open question! We mainly focus on defining the delta learning hypothesis and showing its utility, and view further studying what facets of the delta matter as future work. Still, we are optimistic that our controlled experiments on bolded section generation (`Section 3.1`) offer early evidence that deltas can be targeted to the exact semantic aspects we aim to improve. In other words, we speculate that deltas may be most effective when semantically aligned with gaps in model behavior. One promising future work may curate deltas by synthesizing targeted corruptions along specific axes—e.g., _adding_ unsafe language to generate rejected responses for preference tuning to improve safety.
> >
> > We also manually inspected our best weak preference data (Qwen-3B over 1.5B) to qualitatively understand what semantic deltas were present. Overall, there was no single axis by which Qwen-3B responses are better, but we observed a few interesting deltas for future study:
> > - For verifiable prompts (e.g. math prompts), we find pairs where Qwen-3B responds correctly but Qwen-1.5B does not
> > - For knowledge-seeking prompts, Qwen-3B often gives more detailed responses
> > - For prompts allowing brief answers, we find pairs where Qwen-3B generates a chain-of-thought, while Qwen-1.5B gives just the answer
> >
> > We will add this discussion to our next revision.\
> > \
> > **Questions:**
> >
> > &ensp; **1. How does delta learning perform on safety?**
> >
> > Please see our response above. Thank you!\
> > \
> > &ensp; **2. Is delta learning a complement to SFT, or a substitute?**
> >
> > Great question! Without strong data, SFT often hurts while delta learning can still succeed (`Figure 4`); we are thus optimistic that delta learning can be a more scalable substitute. That said, in a pilot study applying delta learning directly to Llama 3 base pretrained models, we found the trained models often failed to follow instructions. While we did not optimize the delta learning data in this study, the results hint that some SFT may be needed to at least teach basic chat formatting. It remains an exciting open question as to what skills are better instilled with SFT versus delta learning. We will revise to include this discussion to help motivate future work.\
> > \
> > &ensp; **3. What is the degradation with an incorrect delta?**
> >
> > Thanks for the question! To test this, we tuned Tülu-3-8B-SFT (i.e. the base model from our post-training study) on incorrect deltas constructed by swapping the chosen and rejected responses in our weak preference datasets. We try
> > - Qwen-1.5B over Qwen-3B responses
> > - Llama-1B over Llama-3B responses
> >
> > We follow our analysis studies’ hyperparameters (`Appendix D.6`). Results in `Table R2` show large degradation across all tasks under both incorrect deltas (-6.3 points avg. with Qwen, -11.6 with Llama). We will include these results in our next revision.
> >
> > **Table R2: Performance degradations under an incorrect delta**
> > | Model Name  | MMLU | PopQA | MATH | GSM  | AE2  | IFEval | BBH  | DROP | TQA  | HEval | HEval+ | Avg. |
> > |-|-|-|-|-|-|-|-|-|-|-|-|-|
> > | Tülu-3-8B-SFT  | 66.1 | 29.6  | 31.2 | 76.0   | 12.2 | 71.3  | 69.2 | 61.2 | 46.8 | 86.2  | 79.8   | 57.2 |
> > | + Qwen 1.5B over 3B | 60.5 | 29.3  | 23.3 | 69.9 | 4.2  | 42.9   | 66.8 | 60.1 | 43.4 | 83.8  | 76.1   | 50.9 |
> > | + Llama 1B over 3B | 49.0   | 29.2  | 13.5 | 58.8 | 2.8  | 47.3   | 60.3 | 58.5 | 43.4 | 71.0    | 67.6   | 45.6 |
> >
> > &ensp; **4. How do we ensure that preference tuning avoids learning from low-quality responses of weak models?**
> >
> > Great observation! While we empirically show that preference tuning with weak data yields state-of-the-art performance across standard benchmarks (`Table 4`), this does not ensure gains across all model behaviors. Conceptually, delta learning requires a useful delta between the chosen and rejected responses to drive learning. On behaviors where both the chosen and rejected models are equally weak, there is little signal to learn from. Moreover, it is possible for a a given Model X to outperform Model Y in one behavior while underperforming in another. In such cases, no matter how the models are paired, the resulting preference data contains a negative delta in at least one behavior. We observed this directly in the jailbreaking resistance results discussed in response to your Concern 1.
> >
> > Overall, delta learning is not a magic solution and has potential pitfalls. Even so, it shows strong promise, enabling us to greatly simplify open post-training recipes. We are optimistic that future work may curate deltas in a more fine-grained and targeted manner than our simple approach of pairing outputs from two models. More principled curation may maximize the informativeness of each delta and reduce unintended regressions. We will revise our paper with these points—thank you for helping us strengthen our work!\
> > \
> > \
> > We look forward to further discussion!

---

> > > ### Comment · Reviewer_nk1W · 2025-06-09
> > >
> > > Thank you to the authors for this detailed rebuttal.
> > > 1. The added safety evaluation and discussion are very helpful.
> > > 2. The new theoretical results for homogeneous logistic regression are very interesting and do help ground the intuition behind delta learning.
> > > 3. Regarding the semantic properties for deltas, I appreciate the transparency that this remains an open question, and the qualitative observations of the Qwen deltas are interesting. Adding this discussion to the paper will make it clearer to the readers how to think about curating effective deltas.
> > > 4. The incorrect delta experiment (Table R2) is very useful to see.
> > >
> > > Overall, I am happy with how the authors have addressed my concerns. The new results and clarifications meaningfully improve the paper. I have increased my score to 8. Great work by the authors.

---

> > > > ### Author Response · Authors · 2025-06-09
> > > > **Thank you for your time!**
> > > >
> > > > Thank you very much for your reply! We're thrilled that our response addressed your concerns, and appreciate the score increase. We will revise our paper according to the valuable feedback you've provided in review. Thank you for helping us strengthen our work!

---

### Comment · Area_Chair_CjAL · 2025-06-04
**Discussion period**

Dear reviewers,

This is a reminder that the discussion period is currently in progress and will end on June 10th.

I encourage you to read the other reviews as well as the responses by the authors and engage in a discussion.

Thanks a lot!

- AC

---

### Decision · Program_Chairs · 2025-07-07

**Decision:**

Accept

**Comment:**

This paper proposes and validates the delta learning hypothesis, i.e., preference tuning with paired data created from weaker models can improve the performance of a better language model. The authors show that this is indeed possible as long as there is a clear quality delta, hence the name, between the preference pair (i.e., the chosen sample needs to be clearly better than the rejected sample).

The authors first study the hypothesis in somewhat synthetic setups but more importantly show that this approach can generate a DPO preference set that can yield a better Tulu 8B model, compared to the original model which was trained on a  preference set that relied heavily on annotations from GPT-4o.

Overall, reviewers highlighted the novel and interesting contributions and point out that the work is well motivated, well executed, and provides interesting analyses as well as extensive experimental results.

This paper makes an intriguing and significant contribution to language model post-training and I recommend accepting it.